# Retrovirus insertions in host transcripts trigger de novo piRNA immunity

Baptiste Rafanel [ID] [1,2], Liudmila Protsenko [ID] [1,2], Dominik Handler [ID] [1], Julius Brennecke [ID] [1✉] & Kirsten-André Senti [ID] [1✉]

## Abstract

How host organisms adapt their defense systems to newly invading transposable elements remains poorly understood. Here, we show how *Drosophila melanogaster* acquired PIWI-interacting RNA (piRNA)-mediated immunity against the endogenous retrovirus *Tirant*. We uncover two distinct modes of de novo piRNA biogenesis by combining genetics, small RNA profiling, and population genomics. The primary route involves antisense insertions into the *flamenco* cluster, a master locus for transposon control. Unexpectedly, a second, equally potent mechanism arises from antisense *Tirant* insertions within host gene 3′ UTRs. This process requires host gene transcription but is independent of host gene identity. Our findings challenge prevailing models that tie piRNA precursor specification to genomic origin or nuclear RNA processing context. Instead, they reveal a flexible mechanism that turns a critical vulnerability of transposons into an advantage for the host. When transposition occurs into host gene exons, chimeric antisense transcripts are exported to the cytoplasm, inadvertently initiating piRNA production and enabling rapid, adaptive genome defense against new invaders.

**Subject Categories** Evolution & Ecology; Microbiology, Virology & Host Pathogen Interaction; RNA Biology

## Introduction

The PIWI-interacting RNA (piRNA) pathway safeguards germline genome integrity in animals by silencing transposable elements. In this adaptive, small RNA-based defense system, 22–32 nucleotide piRNAs guide PIWI-clade Argonaute proteins to complementary transposon transcripts, directing transcriptional and post-transcriptional silencing mechanisms (Czech et al, 2018; Ozata et al, 2018; Siomi et al, 2011). piRNAs are generated in the cytoplasm from single-stranded transcripts via two interconnected processes: phased biogenesis, in which precursors are cleaved

stepwise in a 5′ to 3′ direction (Han et al, 2015; Homolka et al, 2015; Mohn et al, 2015), and the ping-pong amplification cycle, which enhances piRNA abundance through reciprocal cleavage of complementary sense and antisense transcripts (Brennecke et al, 2007; Gunawardane et al, 2007). Together, these mechanisms constitute a conserved molecular framework for piRNA biogenesis across metazoans (Gainetdinov et al, 2018).

piRNAs predominantly derive from discrete genomic loci known as piRNA clusters. These loci are often enriched in fragmented transposon insertions and are thought to function as molecular memory banks of past transposon invasions (Aravin et al, 2007; Brennecke et al, 2007; Houwing et al, 2007; Malone et al, 2009). In *Drosophila*, for example, the *flamenco* and *77B* clusters are essential for silencing specific transposons (Brennecke et al, 2007; Pelisson et al, 1994; Sarot et al, 2004; Senti et al, 2025), while others may target inactive elements or act redundantly (Gebert et al, 2021). Collectively, these observations have led to the "trap model" of piRNA biogenesis, which proposes that a transposon becomes targetable only after inserting into an active piRNA cluster and thereby donating its sequence to the system (Bergman et al, 2006; Brennecke et al, 2007; Zanni et al, 2013).

Yet this model raises a conceptual problem. Like most cellular RNAs, piRNA precursors are transcribed by RNA polymerase II as single-stranded transcripts (Senti and Brennecke, 2010). What, then, distinguishes an RNA destined for piRNA processing from a conventional mRNA? Current models refer to various criteria, including specific RNA motifs or structures (Homolka et al, 2015; Ishizu et al, 2015), chromatin context or nuclear processing events unique to piRNA precursors (Le Thomas et al, 2014b; Yu et al, 2019; Zhang et al, 2014), and piRNA-guided slicing (Han et al, 2015; Homolka et al, 2015; Mohn et al, 2015). However, the underlying molecular rules that define a piRNA precursor remain largely unclear. Further complicating matters, the genetic and epigenetic features that specify piRNA cluster identity vary across species and cell types (Li et al, 2013; Malone et al, 2009; Mohn et al, 2014; Robine et al, 2009), and isolated transposon insertions outside of clusters can also elicit piRNA production (Baumgartner et al, 2022; Mohn et al, 2014; Shpiz et al, 2014), challenging the universality of the trap model.

Dissecting how the piRNA pathway distinguishes its precursors from other cellular RNAs is challenging in established transposon control settings, where the initial triggers for piRNA production are

[1]Institute of Molecular Biotechnology of the Austrian Academy of Sciences (IMBA), Vienna BioCenter (VBC), Vienna, Austria. [2]Vienna BioCenter PhD Program, Doctoral School of the University of Vienna and Medical University of Vienna, Vienna, Austria. ✉E-mail: julius.brennecke@imba.oeaw.ac.at; senti@imba.oeaw.ac.at

obscured by layers of evolutionary adaptation. In the *Drosophila* germline, one of the most powerful study systems for piRNA biology, piRNA clusters are epigenetically defined and rely on the Rhino-Deadlock-Cutoff complex to generate piRNA precursor transcripts (Brennecke et al, 2008; Josse et al, 2007; Klattenhoff et al, 2009; Le Thomas et al, 2014a; Mohn et al, 2014; Zhang et al, 2014). However, this chromatin-based mechanism is not conserved outside of fruit flies, and the interplay between maternally inherited piRNAs and chromatin context further complicates the identification of general rules underlying piRNA precursor specification.

To address this, we turned to the simplified piRNA pathway that operates in somatic cells of the *Drosophila* ovary. This pathway targets insect endogenous retroviruses (iERVs) of the *Ty3/gypsy* class, which are specifically expressed in somatic follicle cells to form enveloped viral particles that infect the neighboring germline to achieve new insertions in the germline genome (Leblanc et al, 2000; Pelisson et al, 1994; Sarot et al, 2004; Senti et al, 2025). iERVs are generally silenced by somatic piRNA clusters, genomic loci transcribed in ovarian somatic cells, producing abundant iERV antisense piRNAs mediating silencing. Importantly, the somatic piRNA pathway lacks the Rhino-Deadlock-Cutoff system, operates independently of maternally inherited piRNAs, and is therefore free from germline-specific chromatin regulation (Czech et al, 2018; Senti and Brennecke, 2010).

By leveraging natural *Drosophila melanogaster* populations that have recently experienced the horizontal transfer of an active transposon, we set out to dissect how naïve genomes initiate somatic piRNA production de novo, unconfounded by pre-existing immunity or laboratory conditions. Although once thought rare, at least ten such invasions have occurred in *Drosophila melanogaster* over the past two centuries (Bartolome et al, 2009; Pianezza et al, 2025; Scarpa et al, 2025; Scarpa et al, 2024). One such transposon is the iERV *Tirant*, which likely invaded *D. melanogaster* from *D. simulans* in the mid-20th century (Akkouche et al, 2012; Schwarz et al, 2021). Since then, *Tirant* has spread worldwide, offering a unique opportunity to dissect the molecular events that trigger piRNA biogenesis in response to a novel genome invader.

Here, we show that natural *D. melanogaster* populations have evolved piRNA-mediated resistance to *Tirant* via two distinct mechanisms. As predicted by the trap model, single antisense insertions of *Tirant* into the *flamenco* piRNA cluster are sufficient to initiate piRNA production and silence the element. Strikingly, we also uncovered a second, equally potent mechanism: antisense *Tirant* insertions into the 3′ UTRs of host genes trigger robust piRNA production and transposon repression. We show that piRNA biogenesis from these insertions requires host gene transcription, yet is independent of gene identity. These findings challenge the view that piRNA precursor identity is defined by genomic origin or nuclear RNA processing history. Instead, they highlight the importance of antisense transposon sequences as part of chimeric host transcripts as a trigger for de novo piRNA biogenesis. Our results provide direct support for a model proposed in vertebrates and mosquitoes, in which antisense insertions near host gene 3′ ends can serve as entry points for piRNA immunity against endogenous retroviruses, exogenous RNA viruses, and transposable elements (Konstantinidou et al, 2024; Qu et al, 2023; Yu et al, 2025). Together, this work defines a generalizable framework for how small RNA-based immunity evolves in response to transposon invasions.

# Results

## Natural *D. melanogaster* populations evolved diverse piRNA responses against *Tirant*

*Tirant* belongs to the *ZAM* subclade of *Ty3/gypsy*-class iERVs and encodes three open reading frames: *gag*, *pol*, and *env* (Fig. 1A,B) (Akkouche et al, 2012; Marsano et al, 2000; Schwarz et al, 2021; Senti et al, 2025). To investigate how natural *D. melanogaster* populations responded to the reported *Tirant* invasion during the 20th century (Schwarz et al, 2021), we analyzed the founder lines of the Drosophila Synthetic Population Resource (DSPR), a panel of fifteen strains collected from diverse geographic regions between the years 1930 and 1970 (Fig. 1C) (King et al, 2012). High-quality, long-read genome assemblies are available for all available DSPR strains except *B7* (Chakraborty et al, 2019; Chakraborty et al, 2018). Because the original *B1* and *AB8* strains are no longer accessible, we used the *Ber2* and *Sam* strains as proxies, as they were collected at the same respective locations and time points. In addition to the DSPR panel, we analyzed the *iso-1* strain, a mosaic line derived from laboratory stocks of uncertain origin and collection date (Brizuela et al, 1994). This strain forms the basis of the current *D. melanogaster* reference genome (Hoskins et al, 2015).

We first determined the presence of *Tirant* in each strain by comparing their genomes to the *Tirant* consensus sequence. Two strains (*A1/Canton-S* and *AB8/Sam*) lacked *Tirant* entirely, whereas the remaining twelve harbored variable copy numbers (Fig. 1D), often including at least one structurally intact insertion, indicative of ongoing or recent activity. The collection dates of the DSPR strains align with prior work that placed the *Tirant* invasion between 1940 and 1950 (Schwarz et al, 2021). Notably, *iso-1* carries the highest number of *Tirant* insertions among all strains analyzed (Kaminker et al, 2002).

To test whether strains with *Tirant* insertions established corresponding piRNA responses, we sequenced Argonaute-bound small RNAs from dissected ovaries and early embryos (Grentzinger et al, 2020). Ovarian small RNAs encompass both germline and somatic piRNAs, while small RNAs from early embryos serve as a readout of maternally deposited germline piRNAs (Malone et al, 2009). All *Tirant* positive strains, including *iso-1*, expressed *Tirant* piRNAs in ovaries, although their abundance and biogenesis features varied substantially (Fig. 1D).

Two distinct piRNA pathways operate in the *Drosophila* ovary: one in germline cells and one in the surrounding somatic follicle cells (Brennecke et al, 2007; Gunawardane et al, 2007; Han et al, 2015; Klattenhoff et al, 2009; Li et al, 2009; Malone et al, 2009; Mohn et al, 2015; Mohn et al, 2014; Pelisson et al, 2006; Vagin et al, 2006). Germline cells express three PIWI clade Argonaute proteins (Piwi, Aubergine, Ago3), which are loaded via phased (Piwi and Aubergine) and ping-pong piRNA biogenesis (Aubergine and Ago3). This results in mixed populations of sense and antisense piRNAs, displaying a phasing-index and a characteristic 10-nucleotide overlap known as the ping-pong signature. As an example, Fig. EV1A–C illustrates the characteristic features and genomic profiles of piRNAs targeting *Burdock*, a strictly germline-expressed long terminal repeat (LTR) retroelement. *Burdock* piRNAs display robust ping-pong and phasing signatures. Comparable piRNA profiles in early embryonic samples indicate efficient maternal deposition of germline-derived *Burdock* piRNAs into the

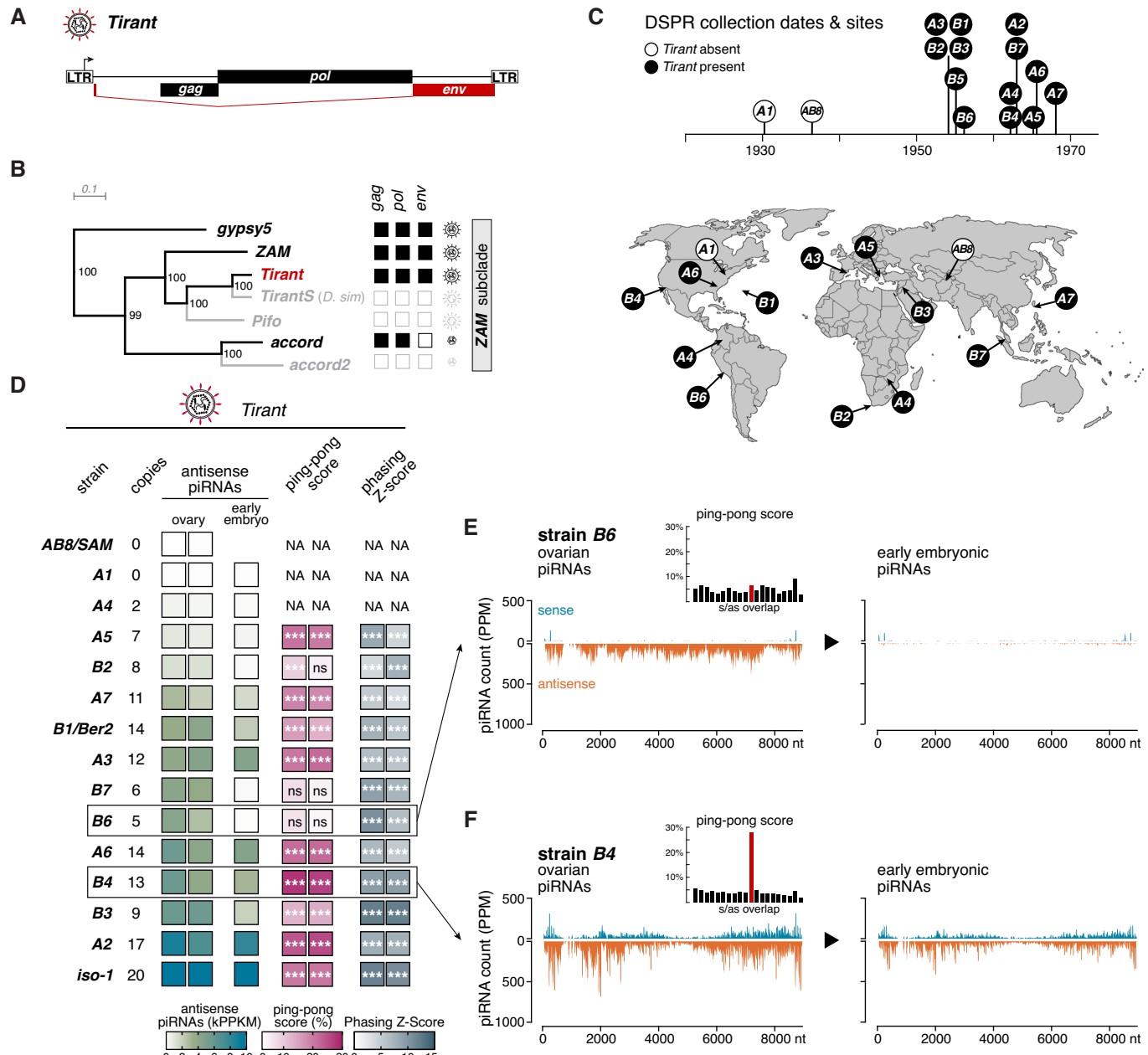

**Figure 1. Natural *D. melanogaster* populations evolved diverse piRNA responses against *Tirant*.**

(A) Schematic of *Tirant*, an iERV, showing its LTRs and ORFs (*gag, pol, env*). (B) Phylogenetic tree of the *ZAM* subclade of the *gypsy/gypsy* clade of iERVs based on full-length Pol sequences (numbers indicate bootstrap values). ORF integrity is shown to the right (black: intact; white: broken). Scale bar: amino acid substitutions/site. (C) Timeline (top) and world map (bottom) of DSPR founder strain collections. *A1* and *AB8/Sam* (white) lack *Tirant*; other strains (black) contain *Tirant* copies (map adapted from Chakraborty et al, 2019; www.outline-world-map.com). (D) Table (left) showing estimated *Tirant* copy numbers from genomic Illumina data and heatmaps (right) of antisense piRNA levels equal or longer than 23 nucleotides (thousand reads per kb, normalized to 1 M miRNAs) mapping perfectly to *Tirant* in ovaries and 0–30 min old embryos, as well as ping-pong and phasing Z-scores (equivalent *P* values: ns equals not significant; * equals <0.05; ** equals <0.01; *** equals <0.001) for ovary samples (NA: insufficient piRNAs for calculation in *A1, A4, AB8/Sam*). Ovarian small RNAs were determined in biological duplicates. (E) Density plots of sense (blue) and antisense (orange) *Tirant*-mapping piRNAs (PPM) along the consensus *Tirant* sequence in *B6* ovaries (left) and early embryos (right). Inset: 5′ overlap histogram of ovarian piRNAs (10-nucleotide ping-pong overlap in red). (F) As in (E), for strain *B4*. See also Fig. EV1 and Dataset EV1.

next generation. In contrast, somatic follicle cells express only Piwi and produce piRNAs exclusively via phased processing of long, single-stranded precursor RNAs. As a result, somatic piRNAs are predominantly antisense, strongly phased, and lack a detectable ping-pong signature. Figure EV1D–F exemplifies these features using piRNAs targeting *gypsy5*, an iERV expressed solely in somatic follicle cells. Consistent with the absence of ping-pong amplification in the soma, *gypsy5* piRNAs show strong phasing but no ping-

pong signal. Moreover, because somatic follicle cells do not contribute piRNAs to the developing embryo, *gypsy5* piRNAs are not maternally deposited.

Analysis of *Tirant* piRNAs revealed that most strains produce both sense and antisense piRNAs, with varying degrees of ping-pong amplification (Fig. 1D–F). Such profiles are consistent with piRNA production in the germline, or in both germline and soma. However, several strains (e.g., *B2, B6,* and *B7*) produced exclusively antisense, phased piRNAs without ping-pong features, indicative of a strictly somatic origin. Supporting this, these strains also lacked maternally inherited *Tirant* piRNAs in early embryos (Fig. 1E).

Altogether, these data reveal that natural *D. melanogaster* populations evolved piRNA responses against the recently invading *Tirant* iERV. However, the variability in the piRNA patterns among fly strains raised the question of which cell types express *Tirant*, and whether both the germline and somatic piRNA pathways contribute to its silencing.

## The somatic piRNA pathway silences *Tirant*

To determine the site of *Tirant* expression in the ovary, we generated transcriptional reporters in which the *Tirant cis*-regulatory sequences (LTR and 5′ UTR) drove either β-Galactosidase (*Tirant-lacZ*) or a GFP:β-Galactosidase fusion (*Tirant-GFP:lacZ*) (Fig. 2A). Each construct was integrated into an *attP* landing site on the second chromosome of a laboratory strain devoid of *Tirant* piRNAs (Fig. 2B). In this piRNA-naïve background, both reporters were strongly and selectively expressed in the ovarian soma, with no detectable germline activity (Fig. 2C,D). However, crossing these reporters to the *iso-1* strain, which produces abundant *Tirant* piRNAs, resulted in silencing, reducing reporter activity to undetectable levels (Fig. 2E–G).

To investigate the expression of endogenous *Tirant* retroviruses, we disrupted the piRNA pathway in flies carrying full-length *Tirant* insertions using transgenic RNA interference (RNAi) (Dietzl et al, 2007; Handler et al, 2013; Ni et al, 2011). This strategy was not feasible in standard laboratory strains as fly lines for tissue-specific RNAi either lacked *Tirant* entirely (*tj*-Gal4 crossed to VDRC dsRNA lines targeting the somatic pathway) or harbored only fragmented insertions (*MTD*-Gal4 crossed to short TRiP hairpin lines for germline knockdown) (Appendix Fig. S1). To overcome these limitations, we generated introgressed lines by crossing *iso-1*, harboring multiple full-length *Tirant* insertions across all major chromosomes, into selected ovarian Gal4 drivers and UAS-RNAi strains. This approach allowed us to interrogate *Tirant* expression in its native genomic context under defined pathway perturbations.

Using RNA fluorescence in situ hybridization (RNA-FISH), we detected no *Tirant* expression in strains lacking insertions (e.g., *A1*) or in unmanipulated *iso-1*, despite harboring ~20 full-length copies. In contrast, disruption of the somatic piRNA pathway in the *iso-1* introgressed background (Fig. 3A; Appendix Fig. S3A) resulted in strong *Tirant* expression across the ovarian soma. Transcripts accumulated in the nuclei and at the apical membranes of follicle cells, with a clear signal also observed in late-stage oocytes, indicating effective soma-to-germline transmission of *Tirant* RNA (Fig. 3A) as previously reported for the related iERV *ZAM* (Senti et al, 2025; Yoth et al, 2023). By contrast, germline piRNA pathway knockdown (Fig. 3B,C; Appendix Fig. S2B,C) did not lead to *Tirant* derepression in either germline or soma (Fig. 3B). Consistent with

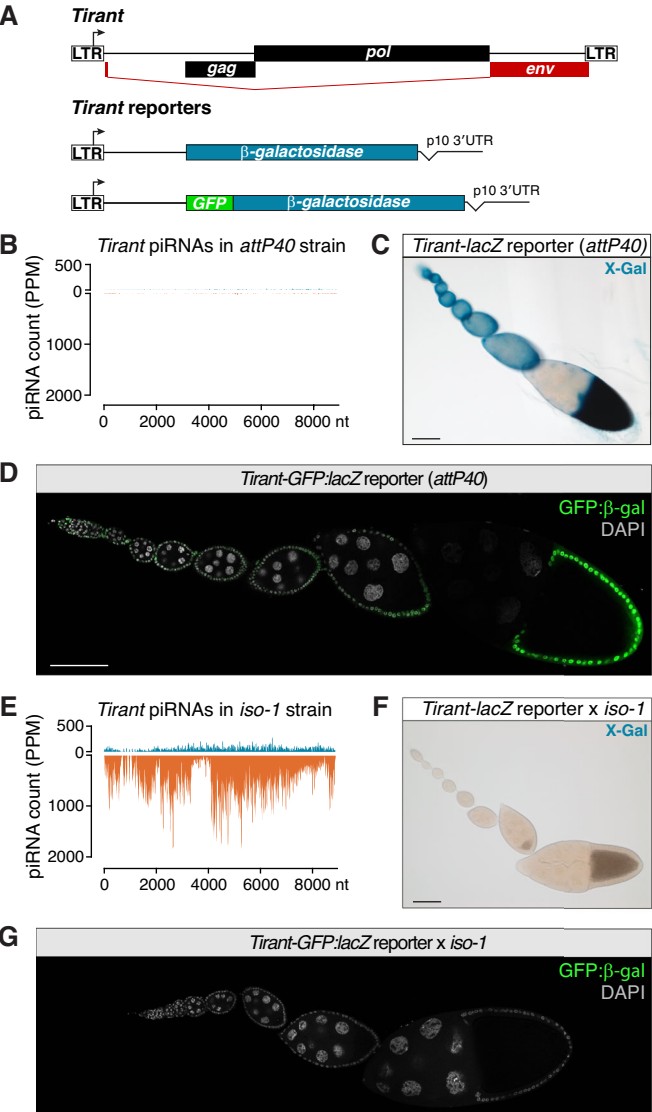

**Figure 2. A *Tirant* reporter is expressed in follicle cells and silenced by piRNAs.**

(A) Schematic depicting the design of the *Tirant-lacZ and GFP:lacZ* reporters. (B) Density plot of *Tirant*-mapping piRNAs (PPM) along the *Tirant* consensus sequence in the *attP40* strain. (C) X-gal staining of an ovariole from transgenic *Tirant-lacZ* reporter flies in the permissive background (*attP40*). Scale bar: 100 μm. (D) GFP signal of the *Tirant-GFP:lacZ* reporter in an ovariole in the permissive genetic background (*attP40*). DNA staining (DAPI) is shown in gray. Scale bar: 100 μm. (E) Density plot of *Tirant*-mapping piRNAs (PPM) along the *Tirant* consensus sequence in the *iso-1* strain. (F) As in (D) but showing an ovariole from the progeny of the cross of transgenic *Tirant-lacZ* reporter females to *iso-1* males. (G) As in (C), but showing an ovariole from the progeny of the cross of transgenic *Tirant-GFP:lacZ* reporter females to *iso-1* males.

these observations, small RNA sequencing of ovaries depleted for germline piRNAs (*MTD*-Gal4-driven *aubergine/Ago3* knockdown) revealed persistent and abundantly produced *Tirant* piRNAs bearing hallmarks of somatic origin, including a pronounced antisense bias, phasing (Z-score: 12.5), and the absence of a ping-pong signature (Fig. 3C; Appendix Fig. S2C; ping-pong Z-score: 0.8).

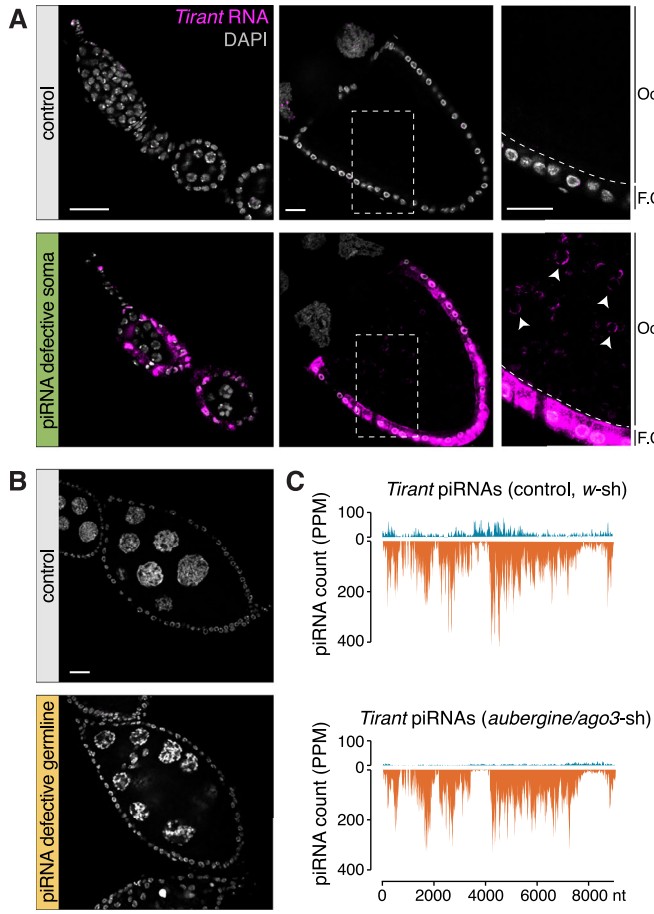

**Figure 3. The somatic piRNA pathway silences the *Tirant* retrovirus.**

(A) RNA-FISH detecting *Tirant* sense transcripts (magenta) in early and late-stage egg chambers of control ovaries (*arrestin2* knockdown; top) or ovaries with defective somatic piRNA pathway (*vreteno* knockdown; bottom). Both genotypes harbor active *Tirant* copies. Right panels are zoomed in from the boxed parts of the middle panels (Oo: oocyte; F.C.: follicle cells, intensity of *Tirant* signal was uniformly increased compared to the left panel to highlight the signal inside the oocyte). Arrowheads show *Tirant* RNA detected in the oocyte. DNA staining (DAPI) is shown in gray. Scale bar: 20 μm. (B) RNA-FISH detecting *Tirant* sense transcripts (magenta) in stage 8 egg chambers with a germline-specific RNAi knockdown of a control gene (*white*; top) or of the germline piRNA pathway (*aubergine* and *ago3*; bottom). Both genotypes harbor active *Tirant* copies. DNA staining (DAPI) is shown in gray. Scale bar: 20 μm. (C) Density plot of *Tirant*-mapping piRNAs (PPM) along the *Tirant* consensus sequence in the strains shown in (B). See also Appendix Figs. S1 and S2.

Taken together, our findings demonstrate that the *Tirant* iERV is intrinsically transcribed in the ovarian soma but is efficiently silenced by the somatic piRNA pathway. Consistent with its capacity to infect the neighboring germline, *Tirant* encodes an intact Env-F glycoprotein, a feature shared with other somatically expressed infectious iERVs. This property likely imposes strong evolutionary pressure on the host to enforce robust silencing of *Tirant* in somatic cells (Marsano et al, 2000; Senti et al, 2025). While somatic piRNAs are sufficient to silence *Tirant*, a contribution of germline *Tirant* piRNAs, which are abundant in several natural strains, to silencing cannot currently be excluded (Akkouche et al, 2013; Yoth et al, 2023).

## Independent antisense insertions in *flamenco* underlie *Tirant* silencing in most natural strains

The DSPR founder panel provided a powerful framework to investigate how natural *D. melanogaster* populations evolved resistance to *Tirant*. These strains represent diverse geographic origins, were largely collected after the estimated *Tirant* invasion, and possess high-quality genome assemblies that allow the accurate mapping of transposon insertions, even in complex heterochromatic regions (Chakraborty et al, 2019; Chakraborty et al, 2018; King et al, 2012).

Given that *Tirant* is transcribed in the ovarian soma and silenced by the somatic piRNA pathway (Fig. 3), we asked whether all natural *Tirant*-carrying strains produce functional *Tirant*-targeting piRNAs in this tissue. To test this, we crossed each strain to the *Tirant-lacZ* reporter line, which lacks *Tirant*-derived piRNAs (Fig. 2B). Reporter silencing in follicle cells served as a direct functional readout for somatic piRNA-mediated repression (Sarot et al, 2004).

As expected, *Tirant*-naïve strains such as *A1* and *AB8/Sam* failed to silence the reporter (Fig. 4A,B). In contrast, all *Tirant*-positive strains repressed the reporter to varying degrees: eight showed complete repression (e.g., *B2*), while five displayed partial (e.g., *A6*) or weak (e.g., *A4*) repression (Fig. 4A,B; Appendix Fig. S3). Reporter repression occurred independently of the parental crossing direction, ruling out a confounding role for maternally inherited piRNAs.

To map the genomic sources of the *Tirant*-repressive piRNAs, we generated hybrid lines in which either the X chromosome or the major autosomes (chromosomes 2 and 3) from each *Tirant*-positive strain were introduced into a balancer background devoid of *Tirant* piRNAs. Crossing these hybrids to the *Tirant-lacZ* reporter revealed the chromosomal origin of functional piRNA source loci. For example, in hybrids from strains *B2* and *A6*, repression occurred only when the DSPR-derived X chromosome was present, pinpointing the piRNA source to the X chromosome (Fig. 4B–D; Appendix Fig. S3).

Notably, in ten of the thirteen repressive strains, functional piRNA source activity mapped exclusively to the X chromosome (Fig. 4D). To identify these loci, we examined all X-linked *Tirant* insertions and analyzed piRNA production from their flanking sequences as a proxy for source activity. Of 45 insertions, six stand-alone euchromatic copies produced weak, bidirectional piRNAs typical of germline piRNA source loci (Appendix Fig. S4; Dataset EV1) (Mohn et al, 2014; Shpiz et al, 2014). Only one euchromatic insertion (in *A3*) produced low-level unistranded piRNAs suggestive of a weak somatic source locus (see below).

In contrast, eight of the ten repressive strains harbored antisense *Tirant* insertions within *flamenco*, the major somatic piRNA cluster located near the pericentromeric heterochromatin of the X chromosome (Brennecke et al, 2007; Pelisson et al, 1994; Sarot et al, 2004; Senti et al, 2025). For the remaining two strains (*B6* and *B7*), a direct assessment was not possible: *B6* lacks a fully assembled *flamenco* locus, and *B7* has no sufficiently assembled genome due to missing long-read data. However, in both cases, RNA-FISH revealed co-localization of *Tirant* antisense transcripts with *flamenco* sense RNA in follicle cell nuclei, mirroring strain *B3*, where a *Tirant* insertion in *flamenco* is confirmed (Fig. 4E). These observations strongly suggest that *B6* and *B7* also carry *Tirant*

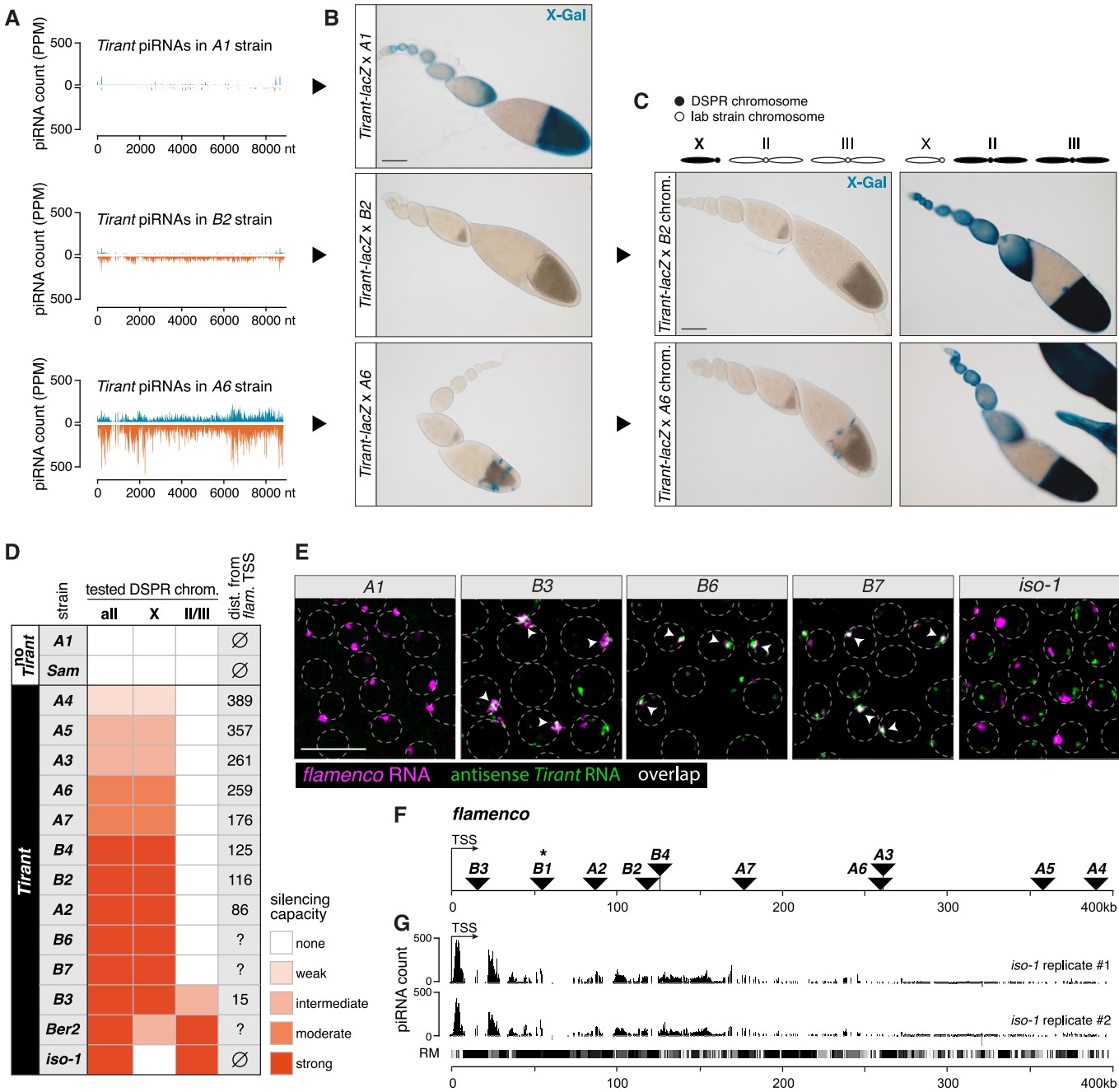

**Figure 4. Independent antisense insertions in *flamenco* underlie *Tirant* silencing in most natural strains.**

(A) Density plots of piRNAs (PPM) mapping to the *Tirant* consensus sequence from the *A1* strain (top), the *B2* strain (middle), or the *A6* strain (bottom). (B) X-gal stainings of ovarioles from the progeny of crosses of transgenic *Tirant-lacZ* reporter females to males from DSPR founder strains shown in (A). Scale bar: 100 μm. (C) X-gal stainings of ovarioles from the progeny of crosses between transgenic *Tirant-lacZ* reporter females and males from hybrid strains harboring the indicated DSPR founder chromosomes of *B2* (top) or *A6* (bottom). Left panels show the tested X chromosomes from DSPR strains, right panels show the autosomes from tested DSPR strains. Scale bar: 100 μm. (D) Summary of the *Tirant-lacZ* reporter silencing capacity of the different DSPR strains (all) and the contribution of the X chromosome or the pair of autosomes for each DSPR strain. The distance between the transcriptional start site (TSS) of *flamenco* and the *Tirant* insertion is shown to the right (distance could not be calculated for the *B6*, *B7*, and *Ber2* strains). (E) RNA-FISH detecting *flamenco* (magenta) and antisense *Tirant* (green) transcripts in somatic follicle cell nuclei of stage 8 egg chambers of the *A1*, *B6*, *B7*, and *iso-1* strains and a stage 10 A egg chamber for the *B3* strain. Somatic follicle cell nuclei circumferences are marked by dashed lines (inferred from DAPI staining). Co-localization of magenta and green signals results in white (marked by arrowheads). Scale bar: 20 μm. (F) Plot showing the localization of *Tirant* insertions (black triangles) in *flamenco* for each DSPR strain relative to the *flamenco* transcriptional start site (TSS). * Indicates that the contribution of the *Tirant flamenco* insertion in the *B1* strain could not be directly assessed. (G) UCSC genome browser screenshot of the *flamenco* locus in *iso-1* (400 kb downstream of the *flamenco* TSS) with ovarian genome-unique mapping piRNAs (PPM) from the *iso-1* strain shown. The RepeatMasker (RM) track is shown at the bottom. See also Appendix Figs. S3, S4, and S7.

insertions in *flamenco*. Notably, with the exception of *A3*, *A6*, and *A7* (likely representing a shared insertion), the *Tirant* insertions in *flamenco* appear to have occurred independently across strains.

*flamenco* is a unistrand piRNA cluster transcribed from a single promoter and spanning several hundred kilobases (Brennecke et al, 2007; Goriaux et al, 2014; Mohn et al, 2014; Senti et al, 2025). Although all identified *Tirant* insertions within *flamenco* were single-copy, silencing efficacy varied widely among strains, ranging from complete repression to minimal effect (Fig. 4C). This variability did not correlate with total ovarian *Tirant* piRNA levels, which are strongly influenced by germline piRNAs (e.g., strains *A3*, *A6*). Instead, we noticed that silencing efficiency was tightly linked to the insertion position relative to the *flamenco* promoter: proximal insertions conferred strong silencing, whereas distal insertions were much less effective (Fig. 4C,F). For instance, strain *A2*, with a *Tirant* insertion 86 kb from the promoter, fully repressed the reporter, while strain *A4*, harboring an insertion ~400 kb downstream, showed minimal repression. Notably, this pattern parallels a drop in piRNA production across the *flamenco* locus, which progressively declines with increasing distance from the promoter, in line with the strong positional effect of integration site on silencing efficacy (Fig. 4G; Dataset EV1). The reduction in piRNA abundance across *flamenco* is most likely attributable to transcriptional drop-off, as suggested by nascent RNA sequencing (Pro-seq) data from Ovarian Somatic Cells (OSCs), in which *flamenco* is highly expressed (Handler and Brennecke, 2025).

During our analysis, we also identified multiple RepeatMasker-annotated degenerate *Tirant*-like fragments within *flamenco*. These correspond to *pifo*, a non-functional *Tirant*-related element largely confined to *flamenco* (Zanni et al, 2013). Since *pifo* fragments were present in all strains, including those incapable of repressing the *Tirant* reporter, we conclude that *pifo*-derived piRNAs, which share ~60% sequence identity with *Tirant*, do not effectively cross-silence *Tirant*.

Together, our findings demonstrate that in most natural strains, *Tirant* repression evolved through independent antisense insertions into *flamenco*, confirming *flamenco* as the primary source of somatic piRNAs. This mirrors observations in *D. simulans*, where the recently invading iERV *Shellder* similarly acquired independent antisense insertions into *flamenco* across multiple strains (Scarpa et al, 2025).

## Antisense insertions into host gene 3′ UTRs provide an alternative silencing route

In three strains, *iso-1*, *B3*, and *Ber2*, *Tirant* silencing mapped partly or entirely to the autosomes (Fig. 4C). In *Ber2*, autosomal activity exceeded that of the X chromosome, while in *iso-1*, silencing was exclusively autosomal. To uncover the origin of these non-*flamenco* piRNA sources, we first focused on *iso-1*, where silencing mapped specifically to chromosome 2, a region lacking any previously described somatic piRNA clusters (Fig. 5A).

Among the *Tirant* insertions on chromosome 2, one stood out: it contained a 752 bp internal deletion coinciding with a sharp drop in antisense piRNA coverage across the element (Fig. 5B). This insertion resides in antisense orientation within the long 3′ UTR isoform of *Fs(2)Ketel (Fs(2)Ket)*, a ubiquitously expressed importin-β gene (Fig. 5C) (Lippai et al, 2000). Consistent with this locus acting as a piRNA source, we detected abundant, unstranded, and phased piRNAs in *iso-1*, but not in other strains such as *A1*,

mapping to the deletion junction and the downstream flanking region (Fig. 5C). The lack of 3′ UTR piRNAs in the *A1* strain indicates that the production of piRNAs in *iso-1* is not an intrinsic feature of the *Fs(2)Ket* 3′ UTR (Robine et al, 2009).

RT-PCR on total ovarian RNA using junction-spanning amplicons confirmed that the *Tirant* insertion is transcribed as part of the *Fs(2)Ket* 3′ UTR (Fig. 5C,D). Since *Fs(2)Ket* is expressed in both soma and germline (Fig. 5E), we used RNA-FISH with intronic probes for *Fs(2)Ket* to determine whether the *Tirant* insertion is transcribed in somatic follicle cells. Indeed, antisense *Tirant* transcripts co-localized with the *Fs(2)Ket* gene locus in follicle cell nuclei but not with *flamenco* (Figs. 5F and 4E), supporting expression as a chimeric host–transposon transcript. Notably, in generating *iso-1* containing RNAi analysis strains, we had also introgressed the *Tirant* bearing allele of *Fs(2)Ket*. The piRNA sequencing of ovaries of the germline control and *aubergine* and *Argonaute3 (Ago3)* RNAi knockdowns revealed that the *Fs(2)Ket* locus indeed produces *Tirant* targeting piRNAs at profound levels in somatic follicle cells (Fig. 3C).

To assess whether *Fs(2)Ket* in *iso-1* was the sole autosomal *Tirant* piRNA source locus, we examined *Ber2* (a proxy for *B1*) and *B3*. Remarkably, *Ber2* also harbors an antisense *Tirant* insertion in the *Fs(2)Ket* 3′ UTR, producing unstranded piRNAs that extend downstream (Fig. EV2A–C). As in *iso-1*, RT-PCR and RNA-FISH confirmed host gene-driven transcription of the insertion (Fig. EV2D,E). Despite occupying the exact same nucleotide position (verified by PCR-sequencing in *Ber2*; Fig. EV2F), the *iso-1* and *Ber2* insertions are most likely independent events, as revealed by distinct SNP profiles reflecting strain-specific *Tirant* polymorphisms (Fig. EV2G).

In *B3*, we identified a third autosomal piRNA source: an antisense *Tirant* insertion in the 3′ UTR of an alternative *cactus* transcript isoform (Fig. EV3A–C). No other chromosome 2 insertions in *B3* produced detectable piRNAs. Although piRNA levels from this locus were modest, consistent with *B3*'s relatively weak autosomal silencing, the piRNA reads contained SNPs unique to this insertion and displayed a downstream unistrand piRNA profile (Fig. EV3B). RT-PCR and RNA-FISH confirmed that this insertion is transcribed as part of *cactus* mRNAs in follicle cells (Fig. EV3D,E).

Finally, we investigated the noncoding RNA *CR44619*, which harbors *Tirant* insertions in two strains (*A3* and *A7*). Strikingly, these insertions occupy the same position in exon 2, but in opposite orientations: antisense in *A3* and sense in *A7*. RT-PCR confirmed their transcription in the respective strains. Notably, only the antisense insertion led to downstream piRNA production, suggesting that piRNA biogenesis in the cytoplasm is influenced by the sequence content of the transcript (Appendix Fig. S5). Consistent with the low expression of *CR44619* in follicle cells, piRNA levels from this locus were low.

Together, these findings uncover an unexpected mechanism of transposon silencing where antisense insertions into host gene 3′ UTRs can act as potent unstranded piRNA sources, bypassing the need for insertions within large piRNA clusters such as *flamenco*.

## A single 3′ UTR insertion is sufficient to silence *Tirant*

To determine whether the *Tirant* insertion in *Fs(2)Ket* is necessary and sufficient for piRNA production and silencing, we first generated a hybrid strain in which the second chromosome from *iso-1* was replaced with balancer chromosomes lacking *Tirant* piRNA sources.

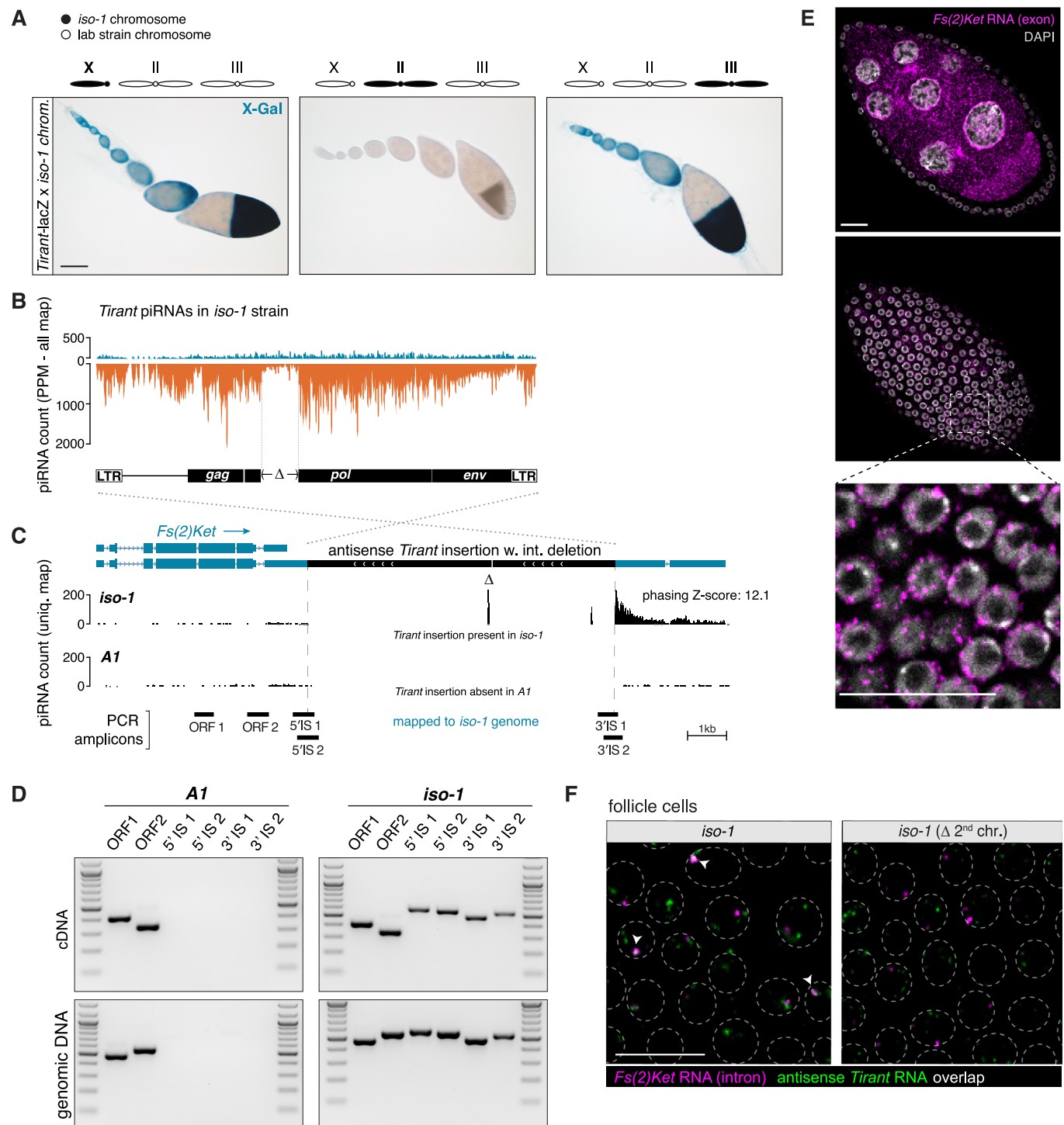

Small RNA sequencing confirmed the loss of piRNA production at the *Fs(2)Ket* locus and a near-complete loss of *Tirant*-targeting piRNAs overall (Fig. 6A; Appendix Fig. S6). As expected, this strain exhibited strong *Tirant* expression as revealed by RNA-FISH (Fig. 6B).

We next asked whether a single genic insertion is sufficient to silence *Tirant* in trans. To this end, we reconstituted the *Fs(2)Ket*

locus from *iso-1* in a *Tirant* naïve strain using a 19 kb genomic BAC (*Pacman*[Fs(2)Ket]) encompassing the entire *Fs(2)Ket* gene and the antisense *Tirant* insertion. This construct was integrated into the *attP2* landing site of a strain lacking *Tirant* piRNAs (Fig. 6A,C). Small RNA sequencing from *Pacman*[Fs(2)Ket] ovaries revealed robust piRNA production from the *Tirant* insertion, including reads spanning the internal deletion (Fig. 6A,D). Downstream of the

**Figure 5.   Antisense insertions into host gene 3′ UTRs provide an alternative silencing route.**

(A) X-gal stainings of ovarioles from the progeny of crosses between transgenic *Tirant-lacZ* reporter females and males from strains harboring either the X (left), 2nd (middle), or 3rd chromosome (right) from *iso-1* as indicated. Scale bar: 100 µm. (B) Density plot of piRNAs (PPM) from the *iso-1* strain mapping to the *Tirant* consensus sequence. Dashed lines mark the region with distinctly lower piRNA coverage, corresponding to the deletion in the *Tirant* insertion in *Fs(2)Ket*. (C) UCSC browser screenshot of the *Fs(2)Ket* locus in the *iso-1* genome. The two tracks show genome-unique piRNAs (in PPM) sequenced from ovaries of *iso-1* (top) or *A1* (bottom). piRNAs mapping uniquely to the internal deletion of *Tirant* are labeled by Δ. Positions of analytic PCR amplicons shown in (E) are indicated at the bottom. Scale bar: 1 kb. Phasing Z-score is restricted to the 3′ UTR piRNAs. (D) RT-PCR on RNA (top) and PCR on genomic DNA (bottom) from the *A1* and *iso-1* strains to detect chimeric transcripts between *Fs(2)Ket* and *Tirant*. Positions of PCR amplicons are shown in panel C. IS: Insertion site. Marker: 100 bp DNA ladder. (E) RNA-FISH detecting *Fs(2)Ket* transcripts (exonic probes, magenta) in germline (top), and somatic cells (middle), with a zoom-in from the boxed part shown below. Images correspond to a stage 7 egg chamber isolated from the *A1* strain. DNA staining (DAPI) is shown in gray. Scale bar: 20 µm. (F) RNA-FISH detecting intronic *Fs(2)Ket* (magenta) and antisense *Tirant* (green) transcripts in follicle cells from stage 8 egg chambers from *iso-1* (left) and in an *iso-1* strain where the second chromosome (containing the *Fs(2)Ket* locus) was replaced by balancer chromosomes (*iso-1(Δ2nd chr.);* right). Circumferences of follicle cell nuclei are marked by dashed lines. Arrowheads indicate foci containing both signals. Scale bar: 20 µm. See also Figs. EV2 and EV3 and Appendix Figs. S5 and S7. Source data are available online for this figure.

insertion, piRNAs displayed a unistranded profile, similar to the endogenous locus in *iso-1* (Fig. 6A).

When crossed to the *Tirant-lacZ* reporter strain, the *Pacman$^{Fs(2)Ket}$* transgene silenced reporter expression completely. Moreover, the transgene was sufficient to silence the full set of endogenous *Tirant* insertions in an *iso-1* background lacking its native second chromosome, as shown by RNA-FISH and RT-qPCR (Fig. 6C–E). RT-PCR and RNA-FISH confirmed transcription of the transgene as a chimeric *Fs(2)Ket–Tirant* transcript (Fig. 6F,G).

The transgene assay enabled us to directly test whether host gene transcription is required for piRNA production and silencing. We inserted a strong cleavage/polyadenylation signal upstream of the *Tirant* insertion in the *Pacman$^{Fs(2)Ket}$* transgene, just downstream of two weak cleavage/polyadenylation sites giving rise to the short *Fs(2)Ket* isoform (*Fs(2)Ket-SV40*, Fig. 7A,B). RT-PCR confirmed efficient transcriptional termination upstream of the *Tirant* sequence (Fig. 7C). Strikingly, flies carrying this modified transgene exhibited a complete loss of *Tirant* silencing, as evidenced by their inability to silence both the *Tirant* reporter and endogenous *Tirant* expression (Fig. 7D). Consistent with this, small RNA sequencing revealed a > 30-fold reduction in *Tirant*-derived piRNAs relative to the original *Pacman$^{Fs(2)Ket}$* strain (Figs. 6D and 7E).

The controlled genetic design of the *Pacman$^{Fs(2)Ket}$* strains enabled us to assess the fitness consequences of uncontrolled *Tirant* expression in the ovarian soma (Fig. EV4A). We generated female flies carrying active *Tirant* insertions from the *iso-1* strain but lacking the antisense *Tirant* insertion at the endogenous *Fs(2)Ket* locus, and introduced one of three transgenic configurations: an empty landing site (no silencing), the *Pacman$^{Fs(2)Ket}$* transgene (silencing), or the *Pacman$^{Fs(2)Ket-SV40}$* transgene (no silencing) (Fig. EV4B). All females laid abundant numbers of eggs; however, the hatching rate of eggs laid by females in which *Tirant* expression was not silenced was moderately (~5%) yet significantly reduced compared with those laid by females in which *Tirant* was silenced. In contrast, introducing the same transgenes into an *iso-1* background that harbors the natural *Fs(2)Ket Tirant* insertion had no detectable effect on fertility (Fig. EV4A).

Collectively, these experiments demonstrate that a single antisense *Tirant* insertion, when transcribed as part of a host gene 3′ UTR, is sufficient to initiate robust piRNA production and drive trans-silencing of *Tirant* in the ovarian soma. Critically, this activity does not rely on *Tirant*-intrinsic promoters but instead depends on transcription from the host gene through the insertion.

## Antisense fragments in the 3′ UTR of an ectopic transgene are sufficient to silence *Tirant*

Our data support a model in which antisense *Tirant* insertions, whether located in *flamenco* or in host gene 3′ UTRs, initiate silencing by generating piRNAs from transcripts that are exported to the cytoplasm, where piRNA biogenesis occurs. Under this model, the essential requirement for piRNA production is the presence of an antisense transposon fragment within a cytoplasmic transcript, rather than the identity of the host gene or a specialized nuclear processing environment.

To directly test this prediction, we generated two minimal UAS-driven transgenes. Each construct encodes GFP followed by an SV40 3′ UTR, into which we inserted a 2 kb *Tirant* fragment encompassing the 5′ UTR and part of *gag*, in either sense or antisense orientation (Fig. 8A). This design allowed controlled expression via the Gal4–UAS system (Brand and Perrimon, 1993) while removing host gene–specific regulatory features present in endogenous piRNA sources.

When expressed by the *tj*-Gal4 driver in ovarian follicle cells of flies lacking endogenous *Tirant* piRNAs but harboring active *Tirant* copies (*iso-1* background without the *Fs(2)Ket-Tirant* insertion), the antisense construct, but not the sense construct, reduced endogenous *Tirant* expression (Fig. 8B). Consistently, only the antisense construct silenced the *Tirant–lacZ* reporter when expressed either throughout the follicle cell epithelium (*tj*-Gal4) (Fig. 8B) or in mosaic GFP-marked clones (Flip-out-Gal4 system) (Fig. 8C). These results demonstrate that antisense orientation alone is sufficient to confer silencing activity in an otherwise neutral transgene context.

Small RNA sequencing from *tj*-Gal4–driven transgenic strains confirmed the production of piRNAs from the inserted *Tirant* fragments (Fig. 8D). In the antisense construct, piRNA production dropped approximately 700 nucleotides into the *Tirant* sequence, likely reflecting premature transcriptional termination caused by a cryptic cleavage and polyadenylation signal within the fragment. This pattern provides a plausible explanation for the weaker silencing efficiency of the UAS transgene relative to the endogenous *Fs(2)Ket-Tirant* source. Notably, we also detected 21 nucleotide siRNAs derived from both strands of the antisense construct (Fig. EV5A). Such siRNAs were not observed in flies expressing *Tirant* piRNAs from the *Fs(2)Ket* transgene and most likely arose from double-stranded RNA formed by pairing between the transgene-derived antisense RNA and *Tirant* sense transcripts due to incomplete silencing. Functional assays

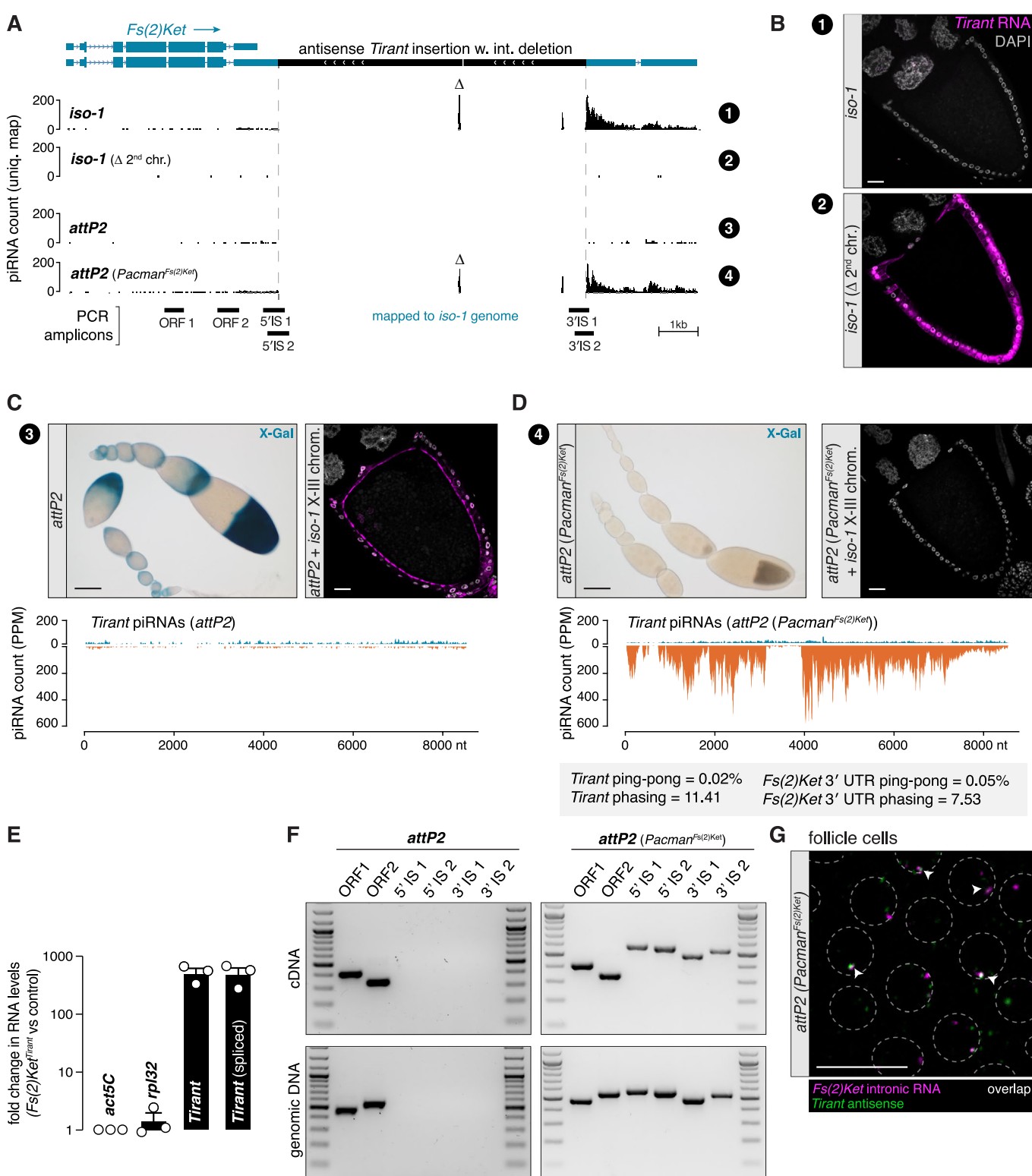

demonstrated that silencing by the antisense UAS transgene requires an intact somatic piRNA pathway: knockdown of the core piRNA pathway component Vreteno in the soma abolished repression of the *Tirant–lacZ* reporter, establishing piRNAs as the functionally relevant silencing species (Fig. 8E). Consistent with this conclusion, nuclear lacZ transcripts were absent in cells expressing the antisense constructs (Fig. EV5B), supporting transcriptional repression by nuclear Piwi–piRNA complexes

◀ **Figure 6.   A single 3′ UTR insertion is sufficient to silence *Tirant*.**

(A) UCSC browser screenshot of the *Fs(2)Ket* locus in the *iso-1* genome. The four tracks show genome-unique piRNAs (in PPM) from ovaries of the *iso-1* strain, an *iso-1* strain with the second chromosome exchanged by balancer chromosomes (*iso-1(Δ2nd chr.*)), a strain carrying an empty *attP2* landing site, and a strain containing a Pacman clone containing the *Fs(2)Ket* locus from *iso-1* in attP2 (*attP2 (Pacman^Fs(2)Ket^*)). piRNAs mapping uniquely to the internal deletion of *Tirant* are labeled by Δ. Relative genomic positions of analytical PCR amplicons shown in panel D are indicated at the bottom. Scale bar: 1 kb. (B) RNA-FISH detecting *Tirant* sense transcripts (magenta) in stage 10 A egg chambers from the *iso-1* strain (top) in which *Tirant* is silenced and in the *iso-1^Δ 2nd chr^* strain (bottom), in which *Tirant* is expressed. DNA (DAPI) is shown in gray. Scale bar: 20 μm. (C) X-Gal staining of an ovariole of the progeny of *Tirant* reporter females crossed to males of the empty *attP2* strain (top left, scale bar:100 μm), and RNA-FISH detecting *Tirant* sense transcripts (magenta) in a stage 10 A egg chamber of a strain carrying the empty *attP2* landing site and the X and 3rd chromosome of the *iso-1* strain (top right, scale bar: 20 μm). Density plot showing ovarian piRNAs mapped to *Tirant* (in PPM) isolated from the *attP2* strain (bottom). (D) As in (C) but in the progeny of *Tirant* reporter females crossed to males carrying the Pacman clone containing the *Fs(2)Ket* locus from *iso-1* (*Pacman^Fs(2)Ket^*) in the *attP2* landing site. Density plot showing ovarian piRNAs mapped to *Tirant* (in PPM) isolated from ovaries carrying the *Pacman^Fs(2)Ket^* clone from *iso-1* in attP2 (bottom). Note the shown ping-pong signature (in %) and phasing Z-scores values for the ovarian piRNAs derived from *Fs(2)Ket* transgenic flies for the *Tirant* insertion itself and the downstream endogenous 3′ UTR of *Fs(2)Ket*. (E) RT-qPCR showing fold change ($\log_{10}$ scale) of *Tirant* and spliced *Tirant env-F* transcripts between ovarian RNA from the strains used for RNA-FISH in (D) versus (C), normalized to *act5C* RNA levels and showing *rpl32* as a housekeeping transcript control ($n = 3$ biological replicates; error bars: standard error of the mean). (F) RT-PCR for *Fs(2)Ket* on ovarian RNA isolated from the empty *attP2* strain (left), and *attP2* (*Pacman^Fs(2)Ket^*) strain (right), and the respective control PCR on genomic DNA. Positions of amplicons indicated in (A). Marker: 100 bp DNA ladder. (G) RNA-FISH detecting *Fs(2)Ket* transcripts (intronic probes; magenta) and antisense *Tirant* transcripts (green) in the *attP2* (*Pacman^Fs(2)Ket^*) strain. Circumferences of follicle cell nuclei are marked by dashed lines. Arrowheads indicate foci with both signals co-localizing. Scale bar: 20 μm. See also Appendix Figs. S6 and S7. Source data are available online for this figure.

rather than post-transcriptional silencing by cytoplasmic Ago2–siRNA complexes.

Together, these experiments demonstrate that embedding an antisense transposon fragment within a cytoplasmic transcript is sufficient to trigger piRNA production and trans-silencing. Although piRNA output from the ectopic transgene is quantitatively lower than that of endogenous sources, the qualitative requirements for silencing are fully recapitulated. Crucially, this activity does not depend on insertion into a piRNA cluster or on a specific host gene environment, supporting the generality of our model.

## Discussion

Transposable elements are typically neutral or deleterious to host fitness, but in rare instances, their insertions can be co-opted for host benefit (Barron et al, 2014). In this study, we describe such a case in the recent invasion of the *Tirant* iERV into *Drosophila melanogaster*. This horizontally acquired transposable element posed an immediate threat to genome integrity, yet specific insertions into host transcriptional units enabled the emergence of an effective piRNA-based immune response. These findings demonstrate how transposition, the very mechanism fueling transposon spread, becomes the Achilles' heel of transposons by eventually generating the transcripts required to silence them.

The central conceptual advance of this work is that cytoplasmic transcripts containing antisense transposon fragments are potent and sufficient triggers of piRNA biogenesis in the ovarian soma. This principle holds across three contexts: natural antisense insertions into the *flamenco* piRNA cluster, antisense insertions into host gene 3′ UTRs, and synthetic transcripts bearing antisense fragments. These findings reveal that piRNA production can occur independently of piRNA-cluster architecture, chromatin context, or host gene identity. Mechanistically, these observations are supported by recent evidence that the DEAD-box ATPase Fs(1)Yb selects antisense ERV transcripts for phased piRNA biogenesis by recognizing their high uridine content, a direct consequence of the

adenosine-rich nature of retrotransposon genomes (Handler et al, 2026). Thus, our work expands the "trap model" of piRNA immunity (Bergman et al, 2006; Brennecke et al, 2007; Zanni et al, 2013): rather than being restricted to specialized clusters, any sufficiently expressed host locus can serve as a functional trap for antisense TE insertions, rapidly converting invasive sequences into sources of silencing piRNAs.

The extended trap model offers a rapid and flexible route to immunity, as a single antisense insertion into a host 3′ UTR can trigger piRNA production and trans silencing of active transposons. At the population level, however, the two silencing strategies differ in their evolutionary dynamics. The *flamenco* cluster resides in pericentromeric heterochromatin, where low recombination favors long-term retention and drift-driven fixation of beneficial insertions. In contrast, 3′ UTR insertions occur in euchromatic regions with high recombination rates and may incur fitness costs by interfering with host gene expression. Consistent with this constraint, both *Fs(2)Ket* and *cactus*, the genic piRNA source loci identified here, harbor *Tirant* insertions within their less abundant alternative 3′ UTR isoforms, suggesting selection for effective silencing while minimizing deleterious effects. Such trade-offs likely explain why genic piRNA source loci remain rare, despite the widespread presence of alternative 3′ UTRs across the genome (Alfonso-Gonzalez et al, 2023; Lee et al, 2022). Importantly, transgenic *Fs(2)Ket* insertions that generate piRNAs confer a mild but significant fitness advantage in the presence of active *Tirant*, and the occurrence of independent *Tirant* insertions in *Fs(2)Ket* and *cactus* in natural isolates indicates positive selection during the *Tirant* invasion. Together, these findings support a model in which genic 3′ UTR insertions act as rapid, adaptive, but transient silencing solutions that precede the emergence of more stable, cluster-based defenses such as *flamenco*.

The broader significance of the extended trap model is underscored by parallels across diverse species. In wild koala populations, a single antisense insertion of the KoRV-A retrovirus into a host gene's 3′ UTR has been linked to piRNA-mediated immunity against this invading virus (Yu et al, 2025; Yu et al, 2019). Similarly, in mammals and mosquitoes, endogenous retroviral and viral insertions near host gene termini produce antisense transcripts that feed into the piRNA pathway (Konstantinidou et al, 2024; Qu et al, 2023). While these systems differ from the

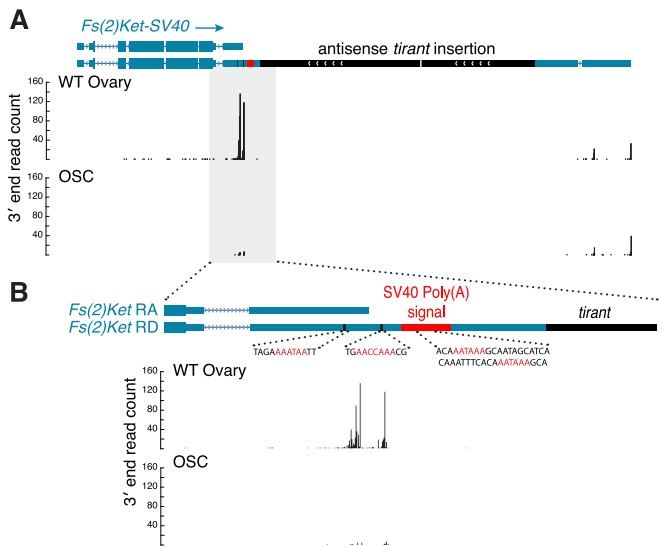

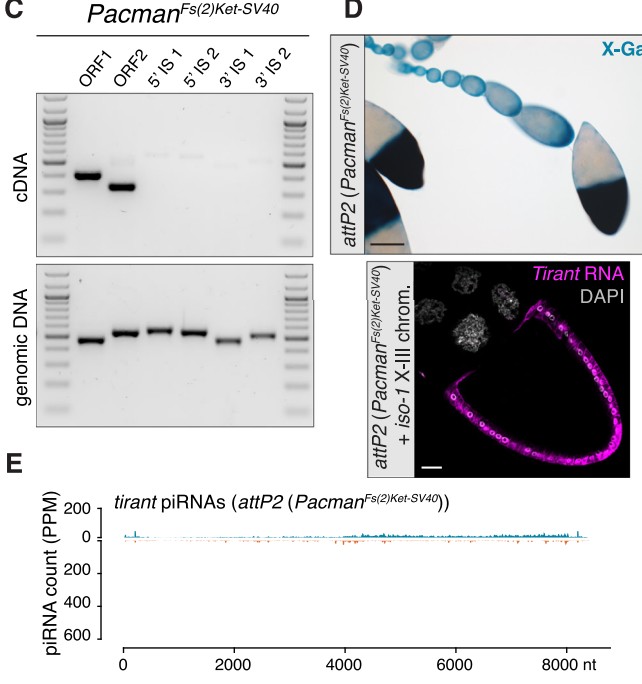

**Figure 7.   piRNA-mediated silencing of *Tirant* requires the *Tirant* insertion to be part of the *Fs(2)Ket* transcript.**

(**A**) Schematic of the *Fs(2)Ket* locus contained in the Pacman clone in which a strong polyadenylation signal from the SV40 3′ UTR was inserted between the end of the short isoform and the *Tirant* insertion (top). Shown below is the 3′ end profile of long-read RNAs from wild-type (WT) ovaries (genotype: tj-Gal4>*arrestin2GD*) and cultured ovarian somatic cells (OSCs), highlighting 3′ ends of the different *Fs(2)Ket* transcripts. (**B**) Zoom-in of the part highlighted in gray in panel A, showing predicted polyadenylation signals from the endogenous *Fs(2)Ket* locus and the inserted *SV40* 3′ UTR. (**C**) RT-PCR from ovarian RNA (top) and PCR from genomic DNA (bottom) from flies harboring the Pacman*Fs(2)Ket-SV40* transgene. Marker: 100 bp DNA ladder. (**D**) X-gal staining of ovarioles from the progeny of *Tirant-lacZ* reporter females crossed to males carrying the *PacmanFs(2)Ket-SV40* transgene. Scale bar: 100μm (top). RNA-FISH detecting *Tirant* sense transcripts (magenta) in a stage 10B egg chamber of a strain carrying an insertion of the *PacmanFs(2)Ket-SV40* in the *attP2* landing site in a genetic background containing the X and the third chromosomes from the *iso-1* strain. DNA staining (DAPI) shown in gray. Scale bar: 20 μm. (**E**) Density plot showing ovarian piRNAs (in PPM) isolated from the *PacmanFs(2)Ket-SV40* strain and mapped to the *Tirant* consensus sequence. See also Fig. EV4. Source data are available online for this figure.

## Methods

### Reagents and tools table

| Reagent/resource | Reference or source | Identifier or catalog number |
| --- | --- | --- |
| **Experimental models** | | |
| *Drosophila melanogaster* stocks | | Dataset EV2 |
| **Recombinant DNA** | | |
| UAS-myrGFP – pJFRC12 | Pfeiffer et al, 2010 | Addgene Plasmid #26222 |
| PACMAN containing the *Fs(2)Ket* locus | Venken et al, 2009 | CH322-190I4 |
| BAC containing a *Tirant* insertion | Hoskins et al, 2000 | BACR48O22 |
| **Antibodies** | | |
| Mouse monoclonal anti-Aubergine 8A9-D7 | Senti et al, 2015 | |
| Rabbit polyclonal anti-Ago3, A14, Rabbit 2404 | Brennecke et al, 2007 | |
| **Oligonucleotides and other sequence-based reagents** | | |
| PCR and RT-qPCR Primer | | Dataset EV2 |
| HCR-RNA-FISH probes | | Dataset EV2 |
| **Chemicals, enzymes, and other reagents** | | |
| Glutaraldehyde | Sigma-Aldrich | G7651-10ML |
| X-gal | Sigma-Aldrich | B4252-1G |
| Formaldehyde | Thermo Fisher Scientific | 28908 |
| Prolong Diamond | Thermo Fisher Scientific | P36961 |
| HCR RNA-FISH (v3.0) Kit | Molecular Instruments | B5 594, B5 647, and B1 647 |
| Kapa Hifi PCR Kits | Roche | 07958838001 |
| TRIzol Reagent | Fisher Scientific | 12034977 |
| DNaseI, RNase-free | Thermo Scientific | MAN0012000 |
| Zymo RCC-25 Kit | Zymo Research | R1017 |
| LunaScript RT Supermix Kit | New England Biolabs (NEB) | E3010L |
| Luna Universal qPCR Master Mix | NEB | E3003L |

*Drosophila* ovarian soma in that their piRNA biogenesis is primarily driven by piRNA-guided slicing—fueling ping-pong amplification and subsequent phased biogenesis—transposon antisense transcripts remain the universal key to effective defense. Recent evidence suggests that this slicer-dependent biogenesis is initiated by naïve piRNAs from active elements; these basal species then steer production toward transposon-silencing piRNAs by cleaving antisense transcripts whose very presence depends on the extended trap model described here (Handler et al, 2026; Shoji and Tomari, 2026). Together, these observations point to a generalizable strategy by which genomes rapidly evolve defenses against invasive genetic elements, highlighting antisense insertions as critical catalysts of piRNA immunity.

| Reagent/resource | Reference or source | Identifier or catalog number |
|---|---|---|
| Monarch Genomic DNA Purification Kit | NEB | T3010L |
| T4 RNA Ligase 1 | NEB | M0204L |
| T4 RNA Ligase 2, truncated KQ | NEB | M0373L |
| ZR small-RNA PAGE Recovery Kit | Zymo Research | R1070 |
| Zymo RCC-5 Kit | Zymo Research | R1016 |
| Zymoclean Gel DNA Recovery Kit | Zymo Research | D4008 |
| **Software** | | |
| Dendroscope (3.8.10) | Huson and Scornavacca, 2012 | |
| MASCE (2.0) | Ranwez et al, 2018 | |
| IQTREE | Trifinopoulos et al, 2016 | |
| UCSC genome Browser | Casper et al, 2026 | |
| Bowtie | Langmead et al, 2009 | |
| GraphPad Prism version 10 | | |
| **Other** | | |
| NovaSeq X Series | Illumina | |
| NextSeq 2000 | Illumina | |
| Aviti | Element Biosciences | |
| Zeiss LSM 880 | Zeiss | |
| Zeiss Imager.Z.2 | Zeiss | |
| Zeiss Axiocam 506 | Zeiss | |

## Fly strains and husbandry

Flies were maintained at 25 °C under a 12-h light/dark cycle. All fly stocks used in this study are listed in Dataset EV2. Tissue-specific knockdowns were performed as described in (Senti et al, 2025). Introgression of *iso-1* chromosomes into the ovarian tissue-specific Gal4 strains and RNAi lines was performed by crossing them to *iso-1* flies carrying different combinations of additional balancer chromosomes. To map the silencing capacity of each DSPR strain or the *iso-1* strain to individual chromosomes, we generated different combinations of hybrid strains by crossing each DSPR strain to a strain containing balancers on the second and third chromosomes, yielding strains that either carry individual strain-specific X-chromosomes, autosomal chromosomes or individual autosomal chromosomes.

### Fertility assay

Ten virgin females of the relevant genotypes were collected and aged with 5 $w^{1118}$ males for two days on apple juice plates supplemented with yeast paste. Flies were then transferred to fresh apple juice plates, and females were left to lay eggs for a period of 4 h, twice a day for two days. The hatching rate was measured as the percentage of hatched eggs from the total number of eggs laid. Hatching rate for each genotype was measured as triplicate, plates with less than 50 eggs laid were excluded from the analysis. Statistical significance was measured with a paired two-sided *t* test (see Dataset EV1 for *P* values).

## Microscopy

### β-gal staining

Chromogenic β-Galactosidase assays were performed as described in (Handler et al, 2013). Briefly, ovaries are dissected into 1× PBS and then fixed for 15 min with 0.5% glutaraldehyde in PBS. Samples are rinsed three times with 1× PBS and washed once with 1× PBS for 10 min. Samples are incubated in staining solution (10 mM sodium phosphate buffer, pH 7.0, 1 mM MgCl$_2$, 150 nM NaCl, 3 mM potassium ferricyanide, 3 mM, potassium ferrocyanide, 0.1% Triton X-100, 0.05% X-gal) for 1 h at 37 °C. Samples were washed with 1× PBS and mounted on a microscope slide with 75% glycerol. Ovaries were imaged on a Zeiss Imager.Z2 with a Zeiss Axiocam 506 color camera and a 10×/0.45 plan-apochromat objective.

### HCR fluorescence in situ hybridization

All RNA-FISH experiments were performed using HCR-FISH (Choi et al, 2018) as described in (Luo et al, 2020). Samples were mounted with Prolong Diamond and imaged on a Zeiss LSM 880 with a 40×/1.4 EC plan-apochromat Oil DIC or a 63×/1.4 plan-apochromat Oil DIC objectives. Probes were designed as in (Glotzer et al, 2022) and are provided in Dataset EV2. Single-channel images of the multiplex RNA-FISH are shown in Appendix Fig. S7.

### Immunofluorescence

Immunofluorescence stainings were performed as described in (ElMaghraby et al, 2019) with mouse monoclonal 8A3-D7 anti-Aubergine (Senti et al, 2015) and rabbit anti-Ago3 (Brennecke et al, 2007) antibodies. Samples were imaged with a Zeiss LSM 880 with a 40×/1.4 EC plan-apochromat Oil DIC objective or a 63×/1.4 plan-apochromat Oil DIC objective.

## Molecular experiments

### Cloning of lacZ reporter and GFP constructs

The *Tirant-lacZ* reporter was cloned as described in (Senti et al, 2025). The *Tirant-GFP:lacZ* reporter was cloned similarly but using a vector containing a *nlsGFP* fused to the *lacZ* gene. The sequence of the LTR and 5′ UTR of *Tirant* were amplified by PCR using a BAC containing an intact copy of *Tirant* as a template (BACR48O22) (Hoskins et al, 2000). Both reporters were integrated into the *attP40* landing site on the second chromosome (Markstein et al, 2008). pJFRC12 (Pfeiffer et al, 2012) was used as a backbone to insert the 2 kb of *Tirant* sequence from the 5′ UTR and beginning of the *gag* gene in the SV40 3′ UTR of the UAS vector. The *Tirant* sequences were amplified by PCR from BACR48O22.

### Double short-hairpin construct cloning

The construct for expressing short-hairpin RNAs against *aubergine* and *ago3* was cloned as described in (Hayashi et al, 2016) using sequences given in Dataset EV2. Efficiency and specificity of the knockdown were tested by Immunofluorescence with antibodies targeting Aubergine and Ago3 (Appendix Fig. S2C).

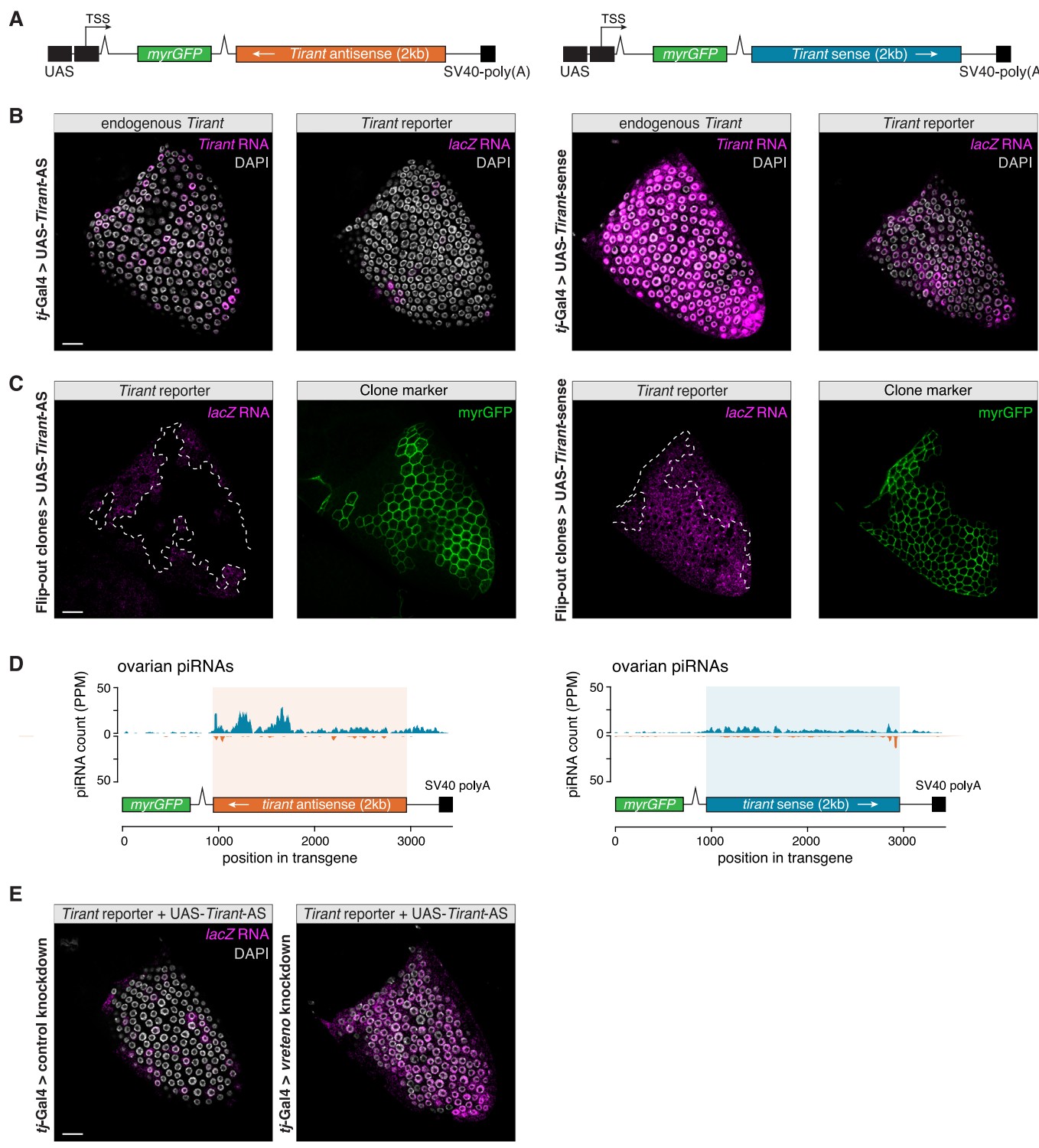

## Fs(2)Ket transgenes

A Pacman containing the whole *Fs(2)Ket* locus from *iso-1* (CH322-190I4) (Venken et al, 2009) was injected into *attP2* to generate the *Pacman^Fs(2)Ket* flies. The *Fs(2)Ket-SV40* Pacman was generated by recombineering as described in (Ejsmont et al, 2011). The SV40 sequence was amplified by PCR from the pJFRC12 vector (Pfeiffer et al, 2012) and fused to the rpsL cassette via fusion PCR.

## RNA extraction for RT-qPCR and RT-PCR

Five pairs of ovaries were dissected and homogenized in TRIzol. RNA was extracted using TRIzol/chloroform, followed by iso-propanol precipitation and washed once with 80% ethanol. The RNA was treated with DnaseI for 30 min and cleaned using a Zymo RRC-25 kit following the manufacturer's instructions. In total, 1 μg of RNA was digested a second time with DnaseI and reverse

**Figure 8.   Antisense fragments in the 3′ UTR of an ectopic transgene are sufficient to silence *Tirant*.**

(A) Schematic of the *Tirant* antisense (left) and sense (right) constructs based on a *UAS-myrGFP* transgene (top). (B) RNA-FISH detection of endogenous *Tirant* transcripts or *Tirant-lacZ* reporter transcripts (both in magenta) in stage 10B egg chambers from the progeny of crosses between flies carrying UAS-myrGFP-*Tirant* antisense (AS) (left) or sense (right) transgenes and the *tj*-Gal4 driver. DNA staining (DAPI) shown in gray. Scale bar: 20 µm. (C) Heatshock-Flippase *actin*-Gal4 Flip-out clones in the somatic epithelium of stage 10B egg chambers expressing the *Tirant-lacZ* reporter (detected by RNA-FISH, magenta) and the UAS-myrGFP-*Tirant* antisense (left) or sense (right) transgenes (myrGFP expression serves as clonal marker). Scale bar: 20 µm. (D) Density plot of piRNAs (reads per 1 million sequenced miRNAs; PPM) sequenced from Argonaute-bound small RNAs extracted from ovaries expressing the indicated UAS-myrGFP-*Tirant* constructs by *tj*-Gal4. (E) RNA-FISH detecting *Tirant-lacZ* transcripts (magenta) in stage 10B egg chambers expressing the myrGFP-*Tirant*-antisense construct and a RNAi construct targeting *arrestin2* (control; left) or *vreteno* (right). DNA staining (DAPI) shown in gray. Scale bar: 20 µm. See also Fig. EV5.

transcribed using the NEB LUNA-RT kit with random primers. qPCR reactions were performed as biological triplicates, with each reaction performed as a technical triplicate. PCR was performed using Taq polymerase, while qPCR used the NEB LUNA qPCR mix (see Dataset EV2 for primer sequences).

### Genomic DNA extraction

Genomic DNA from roughly 20 adult flies of the same genotype was extracted using the NEB Monarch Spin gDNA Extraction Kit following the manufacturer's instructions. Purified genomic DNA was then used directly for PCR.

### Small RNA-seq

Small RNAs bound to Argonaute proteins were isolated using TraPR columns as described in (Grentzinger et al, 2020) and small RNA-seq libraries from ovaries or dechorionated and washed 0–30 min old embryos were generated as described in (Baumgartner et al, 2022) using pre-adenylated DNA linkers containing four random nucleotides at the 5′ end as described in (Jayaprakash et al, 2011).

## Computational analyses

### Sequence alignment and phylogenetics analyses

To place *Tirant* within the phylogeny of the *ZAM* subclade of iERVs in *Drosophila melanogaster*, we compared it with previously described consensus sequences of the ZAM subclade members *gypsy5, ZAM, accord*, and *accord2* (Senti et al, 2025). We further included two *Tirant*-related retroviral consensus sequences, whose fragments are present in the *dm6* reference genome. These are the iERV *Pifo*, initially described in *Drosophila yakuba*, fragments of which are frequently found inserted in *flamenco*, including in *iso-1* (Zanni et al, 2013), and the *Drosophila simulans* iERV *Tirant*S, as partial sequences of this element are harbored in *flamenco* and other heterochromatic sites in *iso-1* (Bargues and Lerat, 2017; Fablet et al, 2006). *Pifo* and to a lesser extent *Tirant*S represent multiple broken sequences previously described as *Tirant*-like in most *Drosophila melanogaster* strains, including those lacking active *Tirant* (Schwarz et al, 2021). For phylogenetical analyses, we annotated the *Pifo*_Dya and *Tirant*S_Dsim consensus sequences for open reading frames and aligned the Pol coding region from all iERV consensus sequences in *Drosophila melanogaster dm6* with MACSE2.0 (Ranwez et al, 2018), followed by IQ-tree phylogenetic tree estimation (Trifinopoulos et al, 2016), and tree display using Dendroscope (Huson and Scornavacca, 2012).

### DSPR genomes

Long read DNA assemblies were collected from (Chakraborty et al, 2019). Annotated genome assembly hubs for the UCSC genome browser were created by aligning reference mRNAs using minimap2 to the published genome assemblies. Repeat annotations were created using RepeatMasker using a curated TE consensus sequence file (attached to the GEO data deposition) (Options: -q -e rmblast -lib TEconsensus.fa).

### Tirant copy number evaluation

For the evaluation of the *Tirant* copy number in the DSPR genomes, the Illumina DNA sequencing data (King et al, 2012) were aligned to the TE-consensus file using bowtie (bowtie --best --strata -f -v 2) (Langmead et al, 2009) and converted to bedgraph files. The median coverage per TE was calculated and normalized to the median coverage of genome unique regions to obtain copy number estimates.

### Analysis of sRNA-seq and RNA-seq

sRNA-seq was analyzed as in (Baumgartner et al, 2022). For embryonic samples, piRNA were first normalized to microRNAs and a ratio of ovary/embryo piRNA levels was calculated for piRNAs exclusively expressed in the germline (*Burdock, F-element, DOC, I-element, mdg3,* and *flea*). PingPong scores were calculated as published in (Hayashi, 2022). RNA-seq and long-read RNA-seq were analyzed as described in (Senti et al, 2025) and (Voichek et al, 2025), respectively. The UCSC Genome Browser (Perez et al, 2025) was used to visualize the aligned piRNAs to the different loci analyzed in this study.

### Mapping of Tirant insertions in DSPR genomes

*Tirant* insertions in the different DSPR founder strains were identified by RepeatMasker (Bao et al, 2015) and extracted as individual sequences from each genome. Insertion coordinates were recorded and compiled into Dataset EV1. For insertions of *Tirant* that occurred within a gene, the orientation of the insertion relative to the transcription of the gene was also recorded. Production of piRNAs from each insertion was assessed by looking at uniquely mapping sRNA around and inside each insertion.

## Data availability

All fly stocks generated for this study are available upon request. The sequencing data have been deposited in the NCBI Gene Expression Omnibus (GEO) under the accession number GSE305718. Raw imaging data is available at BioImage Archive under the accession number S-BIAD2913. The computational code used to analyze the small-RNA-seq data is available at Github (https://github.com/BrenneckeLab/AnnotationPipeline)   (Handler and Brennecke, 2026).

The source data of this paper are collected in the following database record: biostudies:S-SCDT-10_1038-S44318-026-00777-1.

## Peer review information

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

## Acknowledgements

We thank the VBCF and IMBA/IMP/GMI core facilities for support, particularly the NGS facility for sequencing, the BioOptics unit for imaging support, and the VDRC for fly stocks. We thank Peter Duchek and the IMBA Fly & Worm facility for generating transgenic *Drosophila* strains, and Stuart Macdonald, the Drosophila Synthetic Population Resources, and the Bloomington stock centre for sharing fly stocks. Aleksandr Tsarev provided support with sRNA cloning. Members of the Brennecke laboratory and Marianne Yoth gave valuable comments on the manuscript. This work was supported by the Austrian Academy of Sciences, a European Research Council advanced grant (ERC-AdG-101142075; JB), and the Austrian Science Fund FWF grant (P33715-B; KAS). BR is supported by a DOC Fellowship from the Austrian Academy of Science. For the purpose of open access, the authors have applied a CC BY public copyright licence to any Author Accepted Manuscript version arising from this submission.

## Author contributions

**Baptiste Rafanel**: Conceptualization; Data curation; Formal analysis; Validation; Investigation; Visualization; Methodology; Writing—original draft; Writing—review and editing. **Liudmila Protsenko**: Investigation; Writing—review and editing. **Dominik Handler**: Software; Formal analysis; Investigation; Methodology; Writing—review and editing. **Julius Brennecke**: Conceptualization; Resources; Supervision; Funding acquisition; Writing—original draft; Project administration; Writing—review and editing. **Kirsten-André Senti**: Conceptualization; Formal analysis; Supervision; Funding acquisition; Investigation; Methodology; Writing—original draft; Writing—review and editing.

Source data underlying figure panels in this paper may have individual authorship assigned. Where available, figure panel/source data authorship is listed in the following database record: biostudies:S-SCDT-10_1038-S44318-026-00777-1.

## Disclosure and competing interests statement

The authors declare no competing interests. During the preparation of this work, the authors used Abacus AI to polish the writing. After using this tool, the authors reviewed and edited the content as needed and take full responsibility for the content of the publication.

# Expanded View Figures

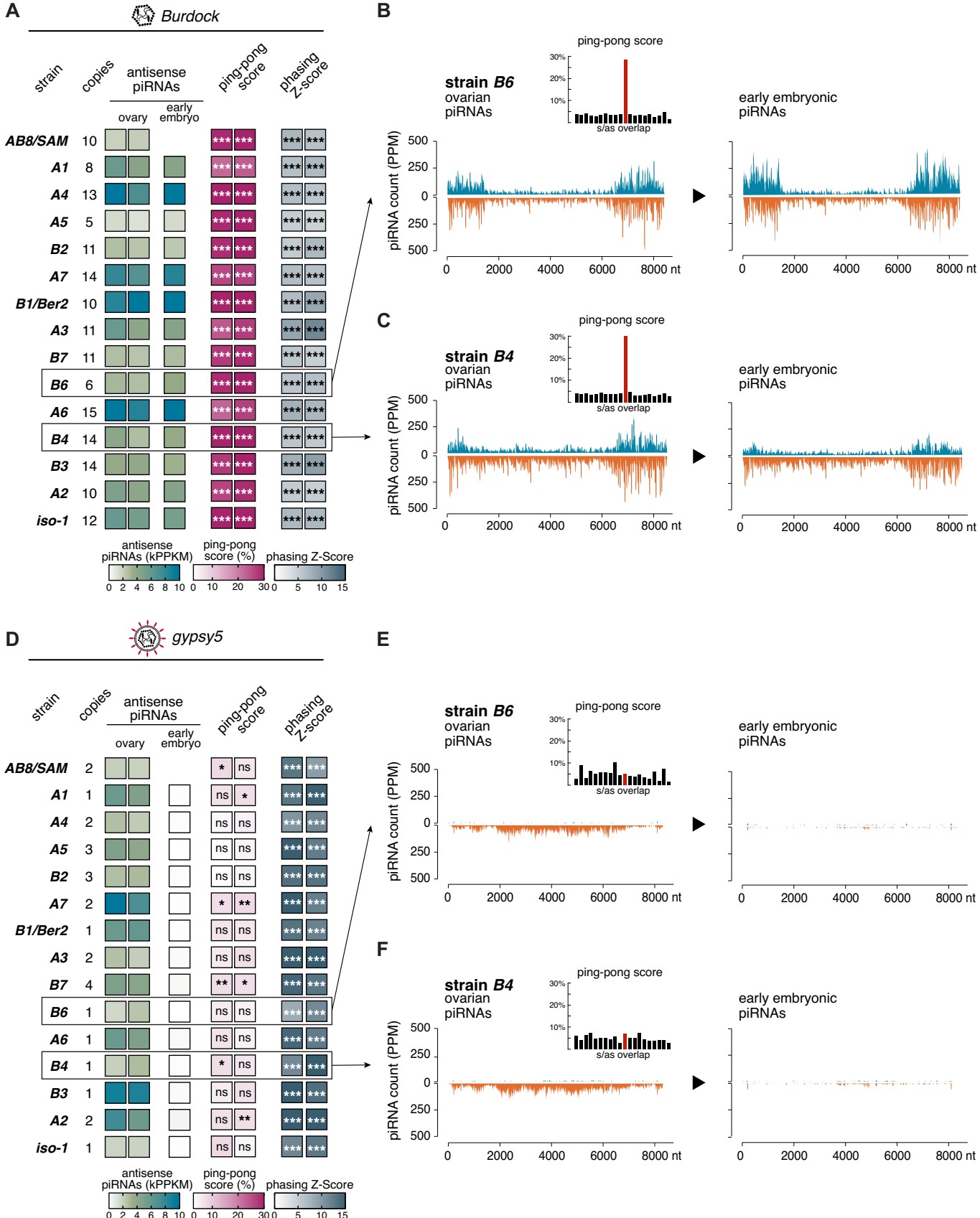

**Figure EV1.** **Characteristics of piRNAs targeting *Burdock* (germline transposon) or *gypsy5* (somatic transposon) in DSPR founder strains.**

(A) Genomic copy numbers of *Burdock* (left) estimated from genomic Illumina data and heatmaps (right) of antisense piRNA levels equal or longer than 23 nucleotides (thousand reads per kb, normalized to 1 million sequenced miRNAs; PPM) mapping antisense to the *Tirant* consensus sequence in total RNA from ovaries and 0–30 min old embryos. Also shown are ping-pong and phasing Z-scores (equivalent *P* values: ns. equals not significant; * equals <0.05; ** equals <0.01; *** equals <0.001) for ovarian piRNAs. All ovarian small RNA data based on biological duplicates. (B) Density plots of sense (positive) and antisense (negative) *Burdock*-mapping piRNAs (PPM) along the *Burdock* consensus sequence in *B6* ovaries (left) and early embryos (right). Inset: 5′ overlap histogram of ovarian piRNAs (10-nucleotide ping-pong overlap in red). (C) As in (B), for strain *B4*. (D) Genomic copy numbers of *gypsy5* (left) estimated from genomic Illumina data and heatmaps (right) of antisense piRNA levels equal or longer than 23 nucleotides (thousand reads per kb, normalized to 1 million sequenced miRNAs; PPM) mapping antisense to the *Tirant* consensus sequence in total RNA from ovaries and 0–30 min old embryos. Also shown are ping-pong and phasing Z-scores (equivalent *P* values: ns. equals not significant; * equals <0.05; ** equals <0.01; ***equals <0.001) for ovarian piRNAs. All ovarian small RNA data based on biological duplicates. (E) Density plots of sense (positive) and antisense (negative) *gypsy5*-mapping piRNAs (PPM) along the *gypsy5* consensus sequence in *B6* ovaries (left) and early embryos (right). Inset: 5′ overlap histogram of ovarian piRNAs (10-nucleotide ping-pong overlap in red). (F) As in (E), for strain *B4*.

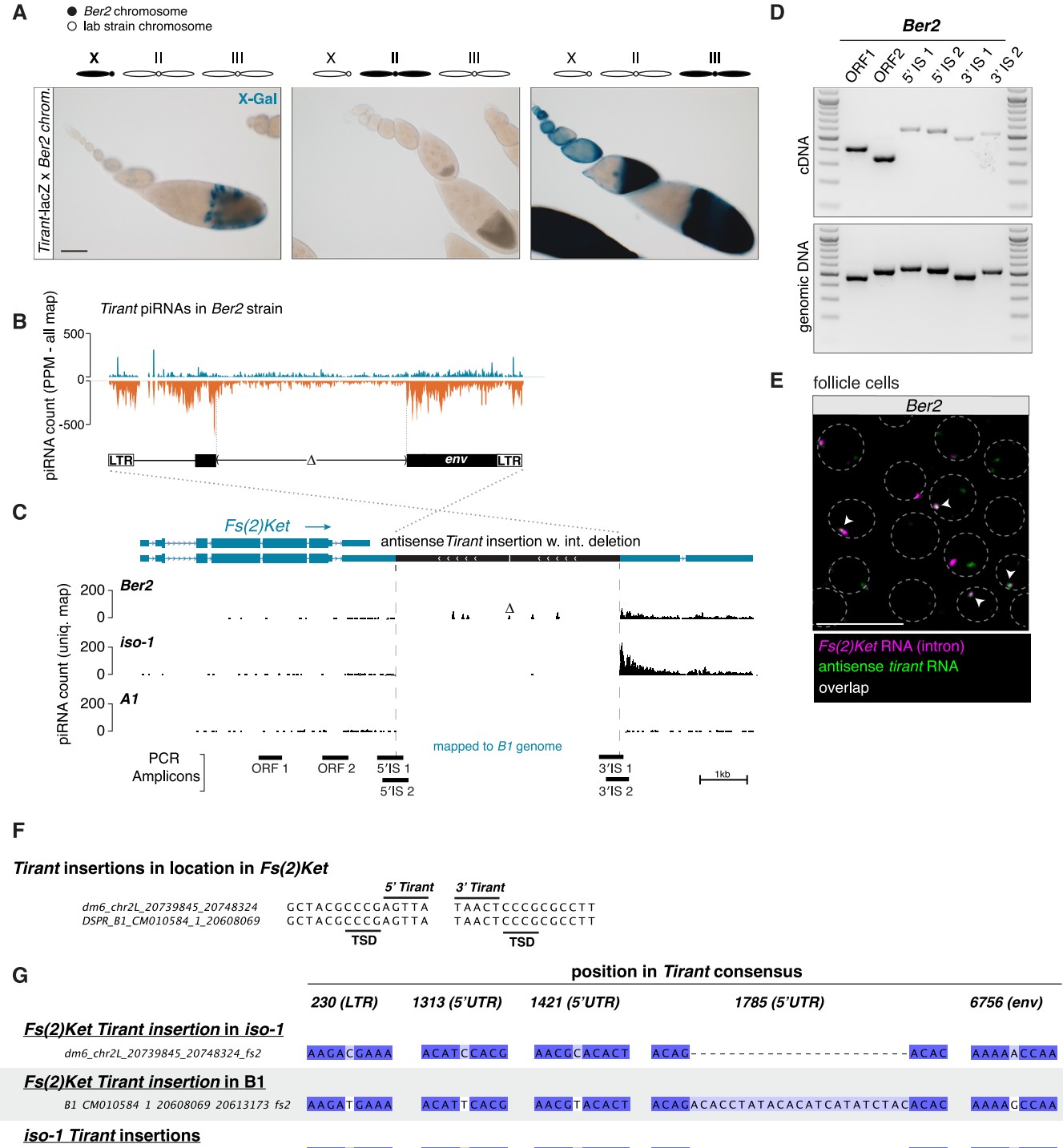

◀ **Figure EV2. A *Tirant* insertion in the *Fs(2)Ket* 3′ UTR is sufficient to produce piRNAs in the *B1/Ber2* strain.**

(**A**) X-gal stainings of ovarioles from the progeny of crosses between transgenic *Tirant-lacZ* reporter females and males harboring either the X (left), 2nd (middle), or 3rd chromosome (right) from the *Ber2* strain, as indicated. Scale bar: 100 μm. (**B**) Density plot of piRNAs (PPM) mapping to the *Tirant* consensus sequence isolated from *Ber2* ovaries. Dashed lines mark the region with distinctly lower piRNA coverage, corresponding to the deletion in the *Tirant* insertion within *Fs(2)Ket*. (**C**) UCSC browser screenshot of the *Fs(2)Ket* locus in the *B1* genome. The three tracks show genome-unique piRNAs (in PPM) that were sequenced from ovaries of *Ber2* (top), *iso-1* (middle), and *A1* (bottom), each mapped to the *B1* long-read genome assembly. piRNAs mapping uniquely to the internal deletion of *Tirant* are labeled by Δ. Relative genomic positions of analytic PCR amplicons shown in panel D are indicated at the bottom. Scale bar: 1 kb. (**D**) RT-PCR on RNA and PCR on genomic DNA from the *Ber2* strain to detect chimeric transcripts between *Fs(2)Ket* and *Tirant*. Positions of PCR amplicons are shown in panel C. Marker: 100 bp DNA ladder. (**E**) RNA-FISH detecting *Fs(2)Ket* (intronic probes; magenta) and antisense *Tirant* (green) transcripts in stage 8 egg chambers of the *Ber2* strain. Circumferences of follicle cell nuclei are marked by dashed lines. Arrowheads indicate foci where both signals co-localize. Scale bar: 20 μm. (**F**) Insertion site sequences of *Tirant* copies in *Fs(2)Ket* in the *iso-1* and *B1* strains. The target site duplications (TSD) flanking the *Tirant* LTRs from both 5′ and 3′ ends are indicated. (**G**) Sequence alignment showing informative parts of the *Tirant* insertions in the *Fs(2)Ket* 3′ UTR present in *iso-1* or *B1* together with a subset of other *Tirant* insertions in these strains. Shown are all SNPs and an insertion in the 5′ UTR that differ between the insertions in *iso-1* and *B1*. Source data are available online for this figure.

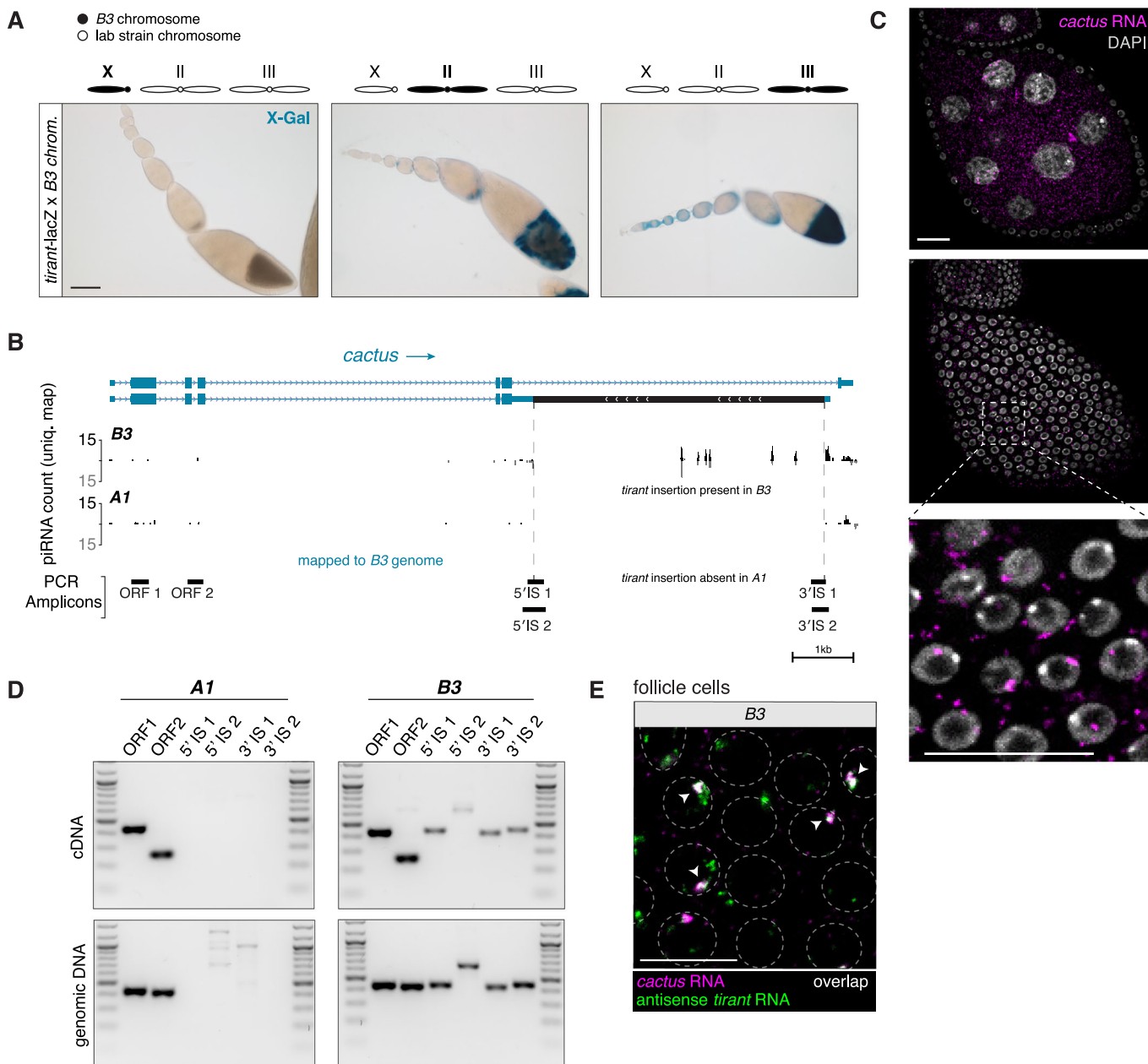

**Figure EV3. A *Tirant* insertion in the *cactus* 3′ UTR is sufficient to produce piRNAs in the *B3* strain.**

(**A**) X-gal stainings of ovarioles from the progeny of crosses between transgenic *Tirant-lacZ* reporter females and males harboring either the X (left), 2nd (middle), or 3rd chromosome (right) from the *B3* strain. Scale bar: 100 μm. (**B**) Genome-unique piRNAs mapping to the *cact* locus in the *B3* genome showing the two annotated *cact* isoforms. piRNAs from *A1* are shown as a negative control. Genomic positions of analytical PCR amplicons shown in panel D are indicated at the bottom. Scale bar: 1 kb. (**C**) RNA-FISH detecting *cact* transcripts (magenta) in the germline (top) or somatic cells (middle) of a Stage 6 egg chamber from the *A1* stain, with zoom-in from the boxed part shown below. DNA staining (DAPI) shown in gray. Scale bar: 20 μm. (**D**) RT-PCR on cDNA and PCR on genomic DNA from the *A1* and *B3* strains to detect chimeric transcripts between *cact* and *Tirant*. Positions of PCR amplicons are shown in (**C**). Marker: 100 bp DNA ladder. (**E**) RNA-FISH detecting *cact* (magenta) and antisense *Tirant* (green) transcripts in follicle cells of ovaries from the *B3* strain. Circumferences of follicle cell nuclei are marked by dashed lines. Arrowheads indicate foci where both signals co-localize. Scale bar: 20 μm. Source data are available online for this figure.

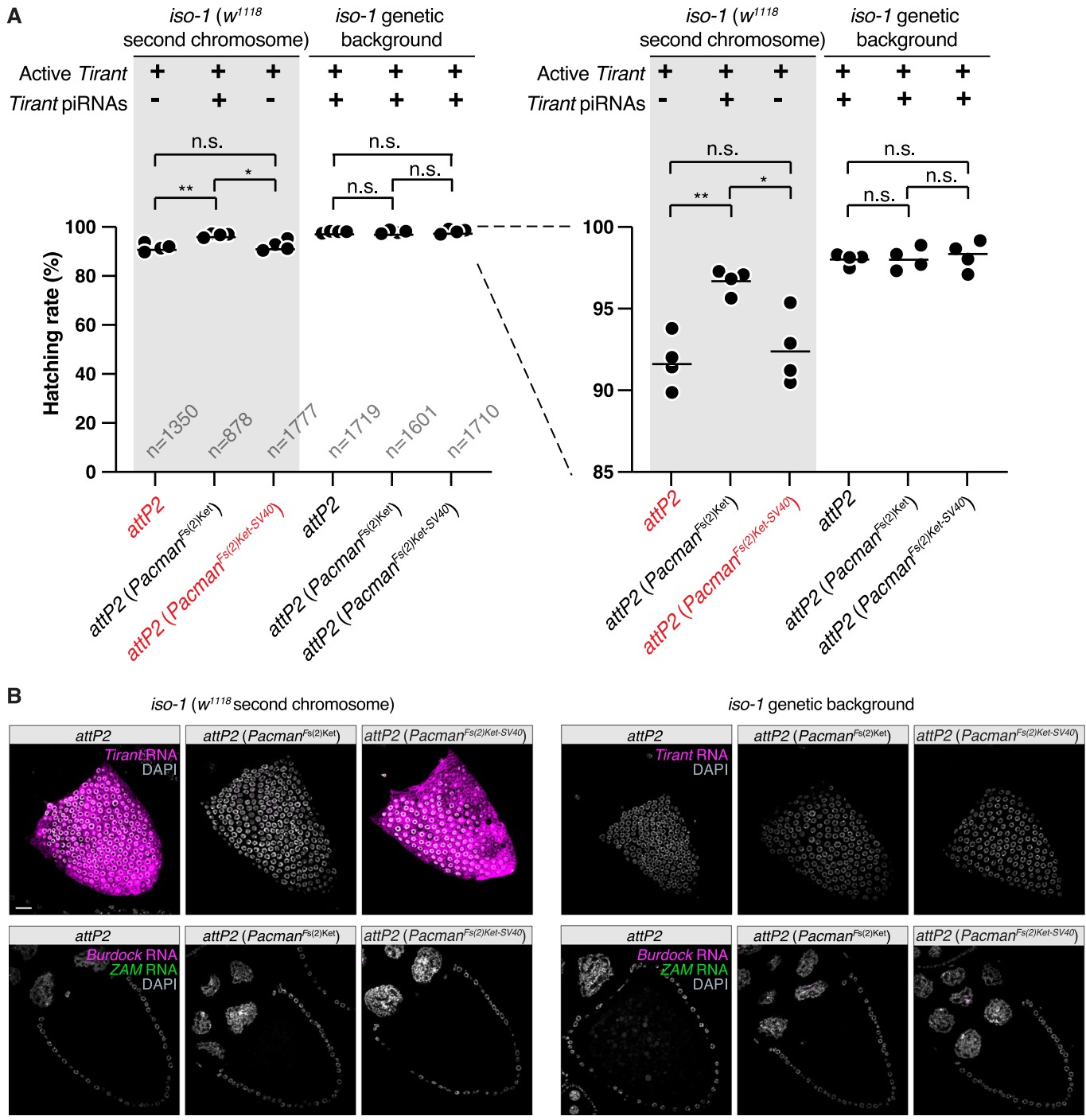

**Figure EV4. The Fs(2)Ket^Tirant transgenic allele confers a significant fertility advantage to females expressing active Tirant.**

(A) Hatching rate of eggs laid by females from crosses between females with indicated genotype (top) and males with indicated genotype (bottom). Each dot represents the mean of three egg collections with the exception of the iso-1;w1118;iso-1 females containing the Fs(2)Ket transgene in attP2 which was measured in duplicate. Genotypes shown in red indicate strains expressing active Tirant (see below). Total egg numbers analyzed are indicated for each genotype. Statistical significance was assessed using an paired two-sided t test (ns, not significant; *P < 0.05; **P < 0.01; ***P < 0.001. In iso-1 (w1118 second chromosome) genetic background, attP2 vs attP2(Pacman^Fs(2)Ket) P = 0.0073, attP2 vs attP2(Pacman^Fs(2)Ket-SV40) P = 0.4144, attP2(Pacman^Fs(2)Ket) vs attP2(Pacman^Fs(2)Ket-SV40) P = 0.0395. In iso-1 genetic background, attP2 vs attP2(Pacman^Fs(2)Ket) P = 0.9046, attP2 vs attP2(Pacman^Fs(2)Ket-SV40) P = 0.7225, attP2(Pacman^Fs(2)Ket) vs attP2(Pacman^Fs(2)Ket-SV40) P = 0.7008, n = 4, corresponding to 4 egg hatching counts for each condition). Bars indicate the mean. (B) RNA-FISH detecting Tirant sense transcripts (magenta – top) and Burdock and ZAM sense transcripts (magenta and green, respectively) in late-stage egg chambers from the strains used in the fertility assay above. DNA staining (DAPI) is shown in gray. Scale bar: 20 µm. Source data are available online for this figure.

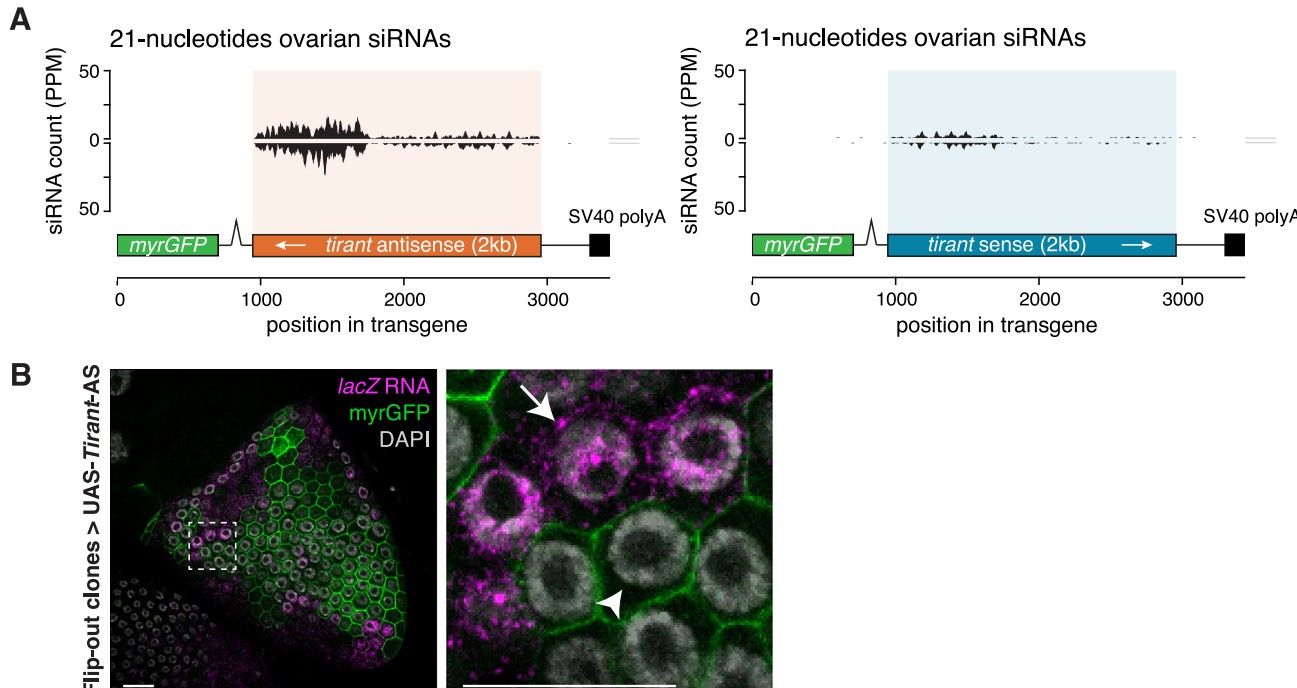

**Figure EV5.   An antisense fragment in the 3′ UTR of a UAS transgene results in siRNA production.**

(A) Density plot of siRNAs (reads per 1 million sequenced miRNAs; PPM) sequenced from Argonaute-bound small RNAs extracted from ovaries expressing the indicated UAS-myrGFP-*Tirant* constructs by *tj*-Gal4. (B) RNA-FISH detecting *lacZ* transcript of the *Tirant* reporter (magenta) in the stage 10B egg chamber used in Fig. 8C (left panel) with a heatshock-induced flip-out clone expressing the UAS-myrGFP-*Tirant* antisense construct. Clones are marked by the presence of GFP (green). Right panel is a zoom-in from the boxed part in the left panel. Arrowhead indicates a cell inside the clone in which no transcripts are detected. The arrow indicates a cell outside the clone in which both cytoplasmic and nuclear transcripts are detected. DNA staining (DAPI) is shown in gray. Scale bar: 20 μm.

