## [Peer Review File · The EMBO Journal]

Retrovirus insertions in host transcripts trigger de novo piRNA immunity

Baptiste Rafanel, Liudmila Protsenko, Dominik Handler, Julius Brennecke, and Kirsten-André Senti

Corresponding authors: Julius Brennecke (julius.brennecke@imba.oeaw.ac.at) , Kirsten-André Senti (senti@imba.oeaw.ac.at)

Review Timeline:

Submission Date:	28th Sep 25
Editorial Decision:	17th Oct 25
Revision Received:	9th Feb 26
Editorial Decision:	3rd Mar 26
Revision Received:	11th Mar 26
Accepted:	24th Mar 26

Editor: Yehu Moran

Transaction Report:

Dear Dr. Brennecke,

Thank you for submitting your manuscript for consideration by the EMBO Journal. It has now been seen by three referees whose comments are shown below. While all three reviewers were enthusiastic about your findings, Referee #1 did raise some more substantial concerns that might require further bioinformatic analyses and experimental work.

Given the referees' positive recommendations, I would like to invite you to submit a revised version of the manuscript, addressing the comments of all three reviewers. I should add that it is EMBO Journal policy to allow only a single round of major revision, and acceptance of your manuscript will therefore depend on the completeness of your responses in this revised version.

I would strongly recommend that after discussing the comments with your co-authors you will prepare a revision plan and share it with me via email. This could help in facilitating a smoother revision process and will improve the acceptance chances.

Thank you for the opportunity to consider your work for publication. I look forward to your revision.

Yours sincerely,

Yehu Moran
Academic Editor
The EMBO Journal

We realize that it is difficult to revise to a specific deadline. In the interest of protecting the conceptual advance provided by the work, we recommend a revision within 3 months (15th Jan 2026). Please discuss the revision progress ahead of this time with the editor if you require more time to complete the revisions.

Referee #1:

Review of manuscript: 'Retrovirus insertions in host transcripts trigger de novo piRNA immunity', Rafanel and co-authors

The authors investigate how *Drosophila melanogaster* acquired piRNA-mediated immunity against the endogenous retrovirus *tirant* and observe two modes of de novo piRNA generation. In doing so, they expand the classical "trap" hypothesis in *Drosophila*, where retroviruses are proposed to insert into pre-existing piRNA clusters, to include a mechanism reminiscent of that described in vertebrates and mosquitoes, in which ERV insertions into 3' untranslated regions give rise to protective piRNAs. The first part of the study maps *tirant*-derived piRNA sources across natural populations (Fig. 1), whereas the remainder focuses on somatic piRNAs in ovarian follicle cells (Figs. 2-8).

This work addresses a fascinating and fundamental question: how new piRNA clusters arise in response to invading retroelements. The combination of population genetics, fly reporter assays, and small RNA sequencing is powerful, and the discovery of genic insertions producing novel piRNAs is compelling. However, several conceptual and experimental gaps currently limit the strength of the conclusions and extend the proposed model beyond the presented data.

Major comments

1. Germline relevance

The exclusive focus on somatic piRNAs is unexpected. Evolutionary adaptation of host defence must occur through the germline, yet no germline piRNA data are provided. Without demonstrating that *tirant* insertions occur in germ cells and yield germline piRNAs it remains unclear how the proposed mechanism could contribute to heritable immunity.

2. Phased piRNA production

The manuscript repeatedly invokes mechanisms of piRNA phasing, yet no quantitative phasing analyses are included. The authors should provide ping-pong and phasing signatures for the key loci and reporter constructs in Figs. 5, 6, and 8, and, if possible, identify the initiating trigger piRNAs. These analyses are essential to substantiate mechanistic claims.

3. Counter-intuitive distance effect

The reported decline in silencing efficiency with increasing distance from the *flamenco* transcription start site (Figs. 2-3) is counter-intuitive, as phased piRNAs should accumulate towards the 3'. The authors should explore whether other parameters (such as different trigger sites, insertion length, orientation, or local chromatin context) correlate with this effect. A more systematic analysis and discussion would clarify this point.

4. Physiological relevance

The biological consequence of *tirant* expression in somatic cells remains insufficiently discussed. Are these flies sterile or otherwise affected?

5. Cluster architecture

The discussion of "canonical" versus "non-canonical" cluster architecture in somatic tissue is vague. The authors should define what constitutes canonical architecture of somatic piRNA clusters and specify how these novel loci differ from established piRNA clusters (for example, in strandedness, length, or chromatin signature).

Overall, this is an original and thought-provoking study that offers valuable insights into the emergence of piRNA sources against newly invading retroelements. The topic is timely, and the work will interest a broad readership. However, to substantiate the broader evolutionary and mechanistic conclusions, the authors need to demonstrate germline relevance, perform rigorous phasing analyses, and better contextualize their findings within established mechanisms of germ cell piRNA biology. With these additions and clarifications, the manuscript would make a strong and impactful contribution to the field.

Referee #2:

This study deals with the question of how small RNA-based defense (piRNAs) originates against transposable elements. It is known that specific genomic loci called piRNA clusters are the producers of most piRNAs in animal gonads, it is also known that 3'UTRs of many genes are processed into piRNAs. Small RNAs from piRNA clusters are shown to orchestrate transposon silencing, while the 3'UTR piRNAs are not shown to function in such a role.

This study demonstrates that even a single antisense insertion of a transposon fragment, into the 3'UTR of an actively expressed gene, is sufficient to produce piRNAs that silence the transposon. It is a nicely executed study that demonstrates the power of fly genetics. I enthusiastically support its publication after some of the below concerns are addressed.

Minor comments:

1. For the RNA-FISH images, eg. fig4.E, as overlap may not be obvious, I would suggest adding extra panels for individual channels.
2. In fig.4A, the plot indicated that piRNA expression is stronger in A6 than B2, however, in fig.4B, it seems that de-silencing happened in A6. In fig.4C, the authors were trying to demonstrate the regularity of silencing and genomic distance, whereas such an effect could be raised by artificial construction (lab strain chromosome).
3. Fig. 8 doesn't clearly present the profile of lacking *Fs(2)Ket-tirant* piRNA source given in context.

Referee #3:

I don't have any serious comments on this manuscript, it was really well done. Two extremely minor comments - RDC is made into an acronym but only used once and I thought that Tirant and Burdock were historically capitalized. Love that the authors included a schematic of collection locations and years for each strain (again, minor comment, but so often it's barely mentioned). Really beautiful paper!

Responses to Reviewer Comments

“Retrovirus insertions in host transcripts trigger de novo piRNA immunity”

Baptiste Rafanel, Liudmila Protsenko, Dominik Handler, Julius Brennecke, Kirsten-André Senti

Reviewer comments are in **blue**.

Responses are in **black**.

General Comments:

We thank all three reviewers for their thoughtful, constructive, and encouraging feedback on our manuscript. Their comments have helped us improve both the clarity and rigor of the presented data and to further strengthen our central conclusions. In the revised manuscript, in addition to addressing the reviewer concerns (see below), we have included a few new experiments that go beyond the specific requests. These additions are:

- Verification of piRNA pathway knockdown efficiency (Appendix Fig S2) for the experiments shown in Fig 3, including:
 - RNA FISH experiments using probes targeting established somatic- (*ZAM*) and germline-expressed (*Burdock*) transposons.
 - Immunofluorescence experiments for Aubergine and Ago3.
- Control experiments demonstrating specificity of *Tirant*-exclusive derepression in Figs 5–7, showing that other well-characterized piRNA-repressed transposable elements are not deregulated (Fig EV4).
- Extended analysis of the UAS-*Tirant* constructs (Fig 8 and Fig EV5), including:
 - Flip-out clone experiments demonstrating the silencing capacity of the UAS-antisense construct compared with neighboring non-expressing cells within the same egg chamber.
 - An additional experiment showing that the silencing activity of the UAS-antisense construct depends on an intact piRNA pathway (Fig 8E).

Point by Point Responses to Reviewer Comments

Reviewer #1:

Review of manuscript: 'Retrovirus insertions in host transcripts trigger de novo piRNA immunity', Rafanel and co-authors

The authors investigate how *Drosophila melanogaster* acquired piRNA-mediated immunity against the endogenous retrovirus *triant* and observe two modes of de novo piRNA generation. In doing so, they expand the classical "trap" hypothesis in *Drosophila*, where retroviruses are proposed to insert into pre-existing piRNA clusters, to include a mechanism reminiscent of that described in vertebrates and mosquitoes, in which ERV insertions into 3' untranslated regions give rise to protective piRNAs. The first part of the study maps *triant*-derived piRNA sources across natural populations (Fig. 1), whereas the remainder focuses on somatic piRNAs in ovarian follicle cells (Figs. 2-8).

This work addresses a fascinating and fundamental question: how new piRNA clusters arise in response to invading retroelements. The combination of population genetics, fly reporter assays, and small RNA sequencing is powerful, and the discovery of genic insertions producing novel piRNAs is compelling. However, several conceptual and experimental gaps currently limit the strength of the conclusions and extend the proposed model beyond the presented data.

We thank Reviewer #1 for their careful and thorough evaluation of our manuscript and for the encouraging assessment of both the biological question and the experimental strategy. We appreciate the reviewer's recognition of the conceptual interest of our study and of the strength of the combined population, genetic, and small RNA approaches.

Regarding the conceptual and experimental concerns raised, several points appeared to reflect ambiguities in our presentation rather than limitations of the data. We have therefore revised the manuscript to improve clarity, better articulate our rationale, and more clearly define the interpretation of our conclusions. In addition, we have performed new experiments and/or analyses in response to the reviewer's suggestions. We believe that these revisions and additions substantially strengthen the manuscript and address the reviewer's concerns.

Major comments

1. Germline relevance

The exclusive focus on somatic piRNAs is unexpected. Evolutionary adaptation of host defence must occur through the germline, yet no germline piRNA data are provided. Without demonstrating that *triant* insertions occur in germ cells and yield germline piRNAs it remains unclear how the proposed mechanism could contribute to heritable immunity.

We agree with the reviewer that heritable adaptation to invading transposable elements generally involves the germline and the production of germline piRNAs. This paradigm applies to most transposable elements, which are cell-autonomous and expressed directly in germline cells such as ovarian nurse cells.

However, the present study focuses on the LTR retrotransposon *Tirant*, which belongs to the class of endogenous retroviruses with a fundamentally distinct biology. *Tirant* is not expressed in the germline. Instead, its promoter activity is restricted to somatic follicle cells of the ovary (Figs 2 and 3). In close analogy to the well-studied endogenous retrovirus *ZAM* (work from the Vaury and Brasset labs), *Tirant* is predicted to form infectious, virus-like particles that invade the adjacent oocyte from the soma with the help of its fusogenic Envelope protein; such germline invasion events are in fact visible in Fig 3A. Consequently, silencing of *Tirant*, as previously demonstrated for *ZAM* and *gypsy*, must occur in somatic follicle cells and is mediated by somatic piRNAs produced predominantly from the *flamenco* piRNA cluster, which itself is expressed exclusively in ovarian somatic cells (Senti et al., 2025; Barckmann et al., 2018; Varoqui et al., 2025).

Despite this strictly somatic expression pattern, natural populations that experienced the *Tirant* invasion harbor stable *Tirant* insertions in their genomes, indicating successful integration into the germline genome. Indeed, prior work on *ZAM* has shown that derepression confined to somatic follicle cells is sufficient to generate new germline insertions (Barckmann et al., 2018; Varoqui et al., 2025). Thus, germline integration occurs through soma-to-germline transmission of viral particles generated in the soma.

We note that several natural strains also produce *Tirant*-derived germline piRNAs, as indicated by the presence of a ping-pong signature and by their maternal deposition in early embryos (Fig 1). While these germline piRNAs may provide an additional layer of protection, we believe they most likely arise secondarily from stochastic engagement of *Tirant* insertions with the germline piRNA pathway. Consistent with this interpretation, disruption of the germline piRNA pathway does not lead to derepression of *Tirant* (Fig 3B). Taken together, our focus on somatic piRNAs reflects the intrinsic biology of *Tirant* and directly addresses the cellular context in which *Tirant* expression, replication, and primary silencing occur. Consistent with this, several natural strains (e.g. *B6*, *B7*, *B2*) express exclusively somatic *Tirant* piRNAs as evident from the lack of ping-pong and maternally deposited *Tirant* piRNAs akin to *gypsy5* piRNAs (Fig EV1 D-F – see also the figure below in the response of reviewer 2's comments). We have revised the text (lines 124-141 and 174-186) to clarify this rationale and to more explicitly state why somatic piRNA immunity is both necessary and sufficient to account for heritable protection against this class of endogenous retroviruses.

2. Phased piRNA production

The manuscript repeatedly invokes mechanisms of piRNA phasing, yet no quantitative phasing analyses are included. The authors should provide ping-pong and phasing signatures for the key loci and reporter constructs in Figs. 5, 6, and 8, and, if possible, identify the initiating trigger piRNAs. These analyses are essential to substantiate mechanistic claims.

We agree with the reviewer that explicit phasing analyses strengthen the mechanistic interpretation of our data. In addition to the ping-pong scores already reported, we have now performed quantitative phasing analyses (using Z-scores) for the natural strains and the experimental strains, where sequencing depth allowed robust calculation. These new analyses

have been added to the revised manuscript (lines 129-147, updated Fig 1D and EV1 for natural strains), and demonstrate robust phasing signatures in *Tirant*-derived piRNA populations, both in natural strains and in strains carrying an experimentally introduced antisense *Tirant* piRNA source (e.g. the *Fs(2)Ket* transgene, Fig 5 and Fig 6).

Regarding the identification of initiating trigger piRNAs, we note that our study focuses on the somatic piRNA pathway in ovarian follicle cells. In this pathway, phased piRNA production is not initiated by PIWI-mediated endonucleolytic cleavage, as the slicer-active PIWI proteins Aubergine and Ago3 are not expressed. Instead, somatic phasing is likely to be initiated by Zucchini-dependent processing without a defined cleavage trigger. Consequently, discrete trigger piRNAs analogous to those in the germline ping-pong pathway cannot be identified in this context.

3. Counter-intuitive distance effect

The reported decline in silencing efficiency with increasing distance from the flamenco transcription start site (Figs. 2-3) is counter-intuitive, as phased piRNAs should accumulate towards the 3'. The authors should explore whether other parameters (such as different trigger sites, insertion length, orientation, or local chromatin context) correlate with this effect. A more systematic analysis and discussion would clarify this point.

We thank the reviewer for raising this point, as it prompted us to better clarify the interpretation of the distance-dependent effects. While it is correct that, under conditions of uniform transcription, phased piRNA production would be expected to stay constant or even accumulate toward the 3' end, the available evidence indicates that transcription across the *flamenco* piRNA cluster is highly non-uniform.

Although all evidence available indicates that *flamenco* is transcribed from a single Pol II promoter, piRNA output progressively declines across the ~700 kb locus. This behavior has been documented both in cultured ovarian somatic cells (OSCs) and in flies. In OSCs, the gradual decrease in piRNA output correlates with reduced transcriptional signal along the cluster, likely caused by widespread transcription termination events through numerous low-efficiency cleavage and polyadenylation sites. The underlying transcriptional data (Pro-seq) supporting this progressive loss of transcriptional output are presented in our recent preprint focusing on OSCs that we now cite in context (Handler & Brennecke. 2025; *bioRxiv*). In the *iso-1* reference strain, the piRNA profiles shown in Fig 4G similarly reveal a pronounced drop in piRNA abundance toward the distal regions of *flamenco*, suggesting that our observations from OSCs hold in vivo (lines 251-254).

Thus, while we agree that phased piRNA production per se should not decline toward the 3' end if transcription were constant, the pronounced decrease in transcriptional output across *flamenco* provides a rationale explanation for the observed reduction in piRNA levels with increasing distance from the transcription start site. This transcriptional gradient directly correlates with the distance-dependent silencing effects described in Fig 4.

Finally, we asked whether other parameters, as suggested by this reviewer, could contribute to the observed silencing differences. Specifically, we tested for correlations with *Tirant* fragment length, insertion orientation, and local chromatin context using available H3K9me3 datasets.

None of these factors showed a convincing association with silencing efficiency, further supporting transcriptional output as the primary determinant.

4. Physiological relevance

The biological consequence of *tirant* expression in somatic cells remains insufficiently discussed. Are these flies sterile or otherwise affected?

We agree with the reviewer that the physiological consequences of *Tirant* expression in somatic cells are an important aspect that warranted further investigation. From our extensive work in flies we know that disruption of transposon silencing in the ovarian soma is less detrimental than disruption of the germline piRNA pathway. For example, depletion of the essential piRNA pathway component Vreano specifically in ovarian somatic cells using *traffic jam*-GAL4 leads to widespread transposon derepression but does not cause full sterility.

Nevertheless, as the reviewer correctly notes, somatic transposon activity would be expected to impose a fitness cost. We therefore directly tested whether expression of active *Tirant* copies affects reproductive fitness. Taking advantage of strains generated in this study, we compared flies with near-identical genetic backgrounds in which *Tirant* is either repressed (functional *Fs(2)Ket* transgene) or active (the same transgene carrying an SV40 stop insertion). We found that eggs laid by flies in which *Tirant* is active exhibit a significantly reduced hatching rate. Although the effect size is modest (approximately a 5% reduction), it is reproducible and statistically significant. This data is now added as Fig EV4 and mentioned in the text (lines 352-361).

While a ~5% reduction in hatching rate under laboratory conditions likely corresponds to a substantial fitness disadvantage in natural populations, these results suggest that the selective pressure to silence somatically expressed transposable elements is lower than that acting on germline-expressed elements such as the *P* element or the *I* element. We have added a corresponding discussion of these findings to the revised manuscript (lines 442-447).

5. Cluster architecture

The discussion of "canonical" versus "non-canonical" cluster architecture in somatic tissue is vague. The authors should define what constitutes canonical architecture of somatic piRNA clusters and specify how these novel loci differ from established piRNA clusters (for example, in strandedness, length, or chromatin signature).

We thank the reviewer for this important comment and agree that our use of the term "canonical piRNA cluster" lacked sufficient definition and context. In the revised manuscript, we have clarified the terminology and avoided the use of "canonical" altogether.

In the context of the somatic piRNA pathway, we now explicitly define somatic piRNA clusters as genomic loci that are transcribed in ovarian somatic cells and give rise to abundant, predominantly antisense piRNAs that mediate transposon silencing (lines 65 and 67). The two best-characterized candidates are *flamenco* and *cluster 77B*, which are large genomic regions

enriched in transposon fragments and fixed across *Drosophila melanogaster* populations (cited in the text at lines 28-36).

Historically, such clusters have been viewed as specialized genomic entities, potentially defined by distinct chromatin signatures or RNA features that selectively route their transcripts into the piRNA biogenesis pathway. A key conclusion of our study is that such specialization is not required for productive piRNA generation. Our data indicate that antisense insertions of retrotransposons into otherwise ordinary, expressed gene loci can be sufficient to trigger robust piRNA production in somatic cells, without the need for pre-existing cluster-like architecture. In this framework, established somatic piRNA clusters such as *flamenco* represent stable, population-fixed major piRNA sources, whereas the genic loci described here are conditional and polymorphic piRNA-producing sites that only acquire this function upon insertion of a transposon.

Overall, this is an original and thought-provoking study that offers valuable insights into the emergence of piRNA sources against newly invading retroelements. The topic is timely, and the work will interest a broad readership. However, to substantiate the broader evolutionary and mechanistic conclusions, the authors need to demonstrate germline relevance, perform rigorous phasing analyses, and better contextualize their findings within established mechanisms of germ cell piRNA biology. With these additions and clarifications, the manuscript would make a strong and impactful contribution to the field.

We thank Reviewer #1 for their thoughtful and constructive assessment of our work and for highlighting both its strengths and the areas requiring clarification or additional support. In response to these comments, we have added new experimental data, including quantitative phasing analyses and fertility assays, and have revised the manuscript to better explain the biological context and mechanistic framework of our findings. We hope that the revised manuscript addresses the reviewer's concerns.

Referee #2:

This study deals with the question of how small RNA-based defense (piRNAs) originates against transposable elements. It is known that specific genomic loci called piRNA clusters are the producers of most piRNAs in animal gonads, it is also known that 3'UTRs of many genes are processed into piRNAs. Small RNAs from piRNA clusters are shown to orchestrate transposon silencing, while the 3'UTR piRNAs are not shown to function in such a role.

This study demonstrates that even a single antisense insertion of a transposon fragment, into the 3'UTR of an actively expressed gene, is sufficient to produce piRNAs that silence the transposon. It is a nicely executed study that demonstrates the power of fly genetics. I enthusiastically support its publication after some of the below concerns are addressed.

Minor comments:

1. For the RNA-FISH images, eg. fig4.E, as overlap may not obvious, I would suggest adding extra panels for individual channel.

We agree with the reviewer that displaying the individual channels improves the clarity of the RNA-FISH data. We have therefore added a supplemental Fig (Appendix Fig S7) showing the separate channels corresponding to Fig 4E, as well as for the RNA-FISH in Figs 5F, 6G, EV2E, and EV3E.

2. In fig.4A, the plot indicated that piRNA expresses stronger in A6 than B2, however, in fig.4B, it seems that de-silencing happened in A6. In fig.4C, the authors were trying to demonstrate the regularity of silencing and genomic distance, whereas such effect could be raised by artificial construction (lab strain chromosome).

We agree that—when considering total ovary piRNA levels—the apparent relationship between piRNA abundance and *Tirant* silencing in strains *A6* and *B2* can appear inconsistent. However, *Tirant* silencing occurs exclusively in the ovarian soma, but the dominant fraction of piRNAs present in whole ovary samples originates from the germline and thus do not contribute to somatic *Tirant* silencing.

The strains *A6*, *A3*, and *A5* produce substantial amounts of germline piRNAs as seen by the strong maternal inheritance of *Tirant* piRNAs cloned from early embryos. This is also supported by the strong ping-pong signature of *Tirant* piRNAs in both ovaries and early embryos (Fig 1D). These germline piRNAs inflate total ovary piRNA counts but are irrelevant for silencing in follicle cells. Considering that the *A3*, *A5*, and *A6* strains and their respective X chromosomes fail to completely silence the *Tirant-lacZ* reporter in somatic follicle cells, somatic *Tirant* piRNA levels must be low, consistent with our proposal that the respective *flamenco* insertions are promoter-distal.

In contrast to this, *Tirant* piRNAs in total ovarian samples from the *B2*, *B6*, and *B7* strains lack ping pong signatures. Moreover, all three strains do not show any maternal deposition of *Tirant* piRNAs (early embryo samples). The ovarian piRNA levels in these three strains therefore correspond directly to the somatic pool, consistent with all strains being strong

silencers of the reporter. We summarized the corresponding data for these six strains in the Fig panel below.

Taken together, whole-ovary piRNA levels are not a reliable predictor of *Tirant* repression in the soma. Instead, our data indicate that the position of the *Tirant* insertion within the *flamenco* piRNA cluster, and specifically its distance from the transcription start site, is a much better predictor of somatic silencing strength. Consistent with this model, transcriptional profiling using Pro-seq in OSCs (as detailed in response to reviewer #1, comment 3) shows that transcription across the *flamenco* cluster drops >10-fold at a distance of 150kb from the promoter. Thus, the observed strength of *Tirant* reporter silencing in vivo is in close agreement

with the transcriptional pattern at the *flamenco* cluster (Handler and Brennecke, 2025). This is mentioned now in the text (lines 251-254)

Regarding the concern that the observed distance effect could arise from artificial chromosomal combinations in laboratory strains, we cannot formally exclude minor influences of the introgressed lab chromosomes. However, the lab strain itself does not produce *Tirant*-derived piRNAs, and the introgression strategy therefore allows us to isolate and test the silencing capacity of individual chromosomes from natural strains. Importantly, the silencing phenotypes observed in the hybrid strains closely match those of the corresponding original strains, as shown in Appendix Fig S3. This strong concordance supports the conclusion that the observed effects reflect intrinsic properties of the natural chromosomes rather than artifacts of laboratory strain construction.

3. Fig. 8 doesn't clearly present the profile of lacking *Fs(2)Ket*-*tirant* piRNA source given in context.

We have sequenced small-RNA from the lines shown in Fig 8 and confirmed that there are no piRNAs produced from the *Fs(2)Ket* locus, consistent with the absence of a *Tirant* insertion in the 3'UTR of *Fs(2)Ket* in these strains. We have attached a genome browser snapshot of the *Fs(2)Ket* locus with piRNAs from the strains used in Fig 8 in the panel below.

Referee #3:

I don't have any serious comments on this manuscript, it was really well done. Two extremely minor comments - RDC is made into an acronym but only used once and I thought that *Tirant* and *Burdock* were historically capitalized. Love that the authors included a schematic of collection locations and years for each strain (again, minor comment, but so often its barely mentioned). Really beautiful paper!

We thank Reviewer #3 for the positive assessment and for the helpful minor suggestions. In response, we have adjusted the nomenclature so that *Tirant* and *Burdock* are capitalized consistently throughout the manuscript, and we have removed the RDC acronym, as it was only used once.

Dear Dr. Brennecke,

As you will see below, the three original referees are satisfied with your revision and remain excited about your results. Yet, Referee #1 makes a justified comment regarding code availability that I am confident you can easily address. Additionally, there are more technical comments from our editorial assistance team that require your attention.

When preparing your letter of response to the referee comments, please bear in mind that this will form part of the Review Process File, and will therefore be available online to the community. For more details on our Transparent Editorial Process, please review our Editorial Policies page: <https://link.springer.com/partners/embo-press/editorial-policies>

Please note that you do not need to address the editorial assistance team comments in your response letter, but correcting the issues they have flagged is mandatory.

We generally allow three months as standard revision time. As a matter of policy, competing manuscripts published during this period will not negatively impact on our assessment of the conceptual advance presented by your study. Yet, I am confident you can fix the remaining issues quite quickly.

Thank you for the opportunity to consider your work for publication. I look forward to your revision so we can proceed to formal acceptance.

Yours sincerely,

Yehu Moran
Academic Editor
The EMBO Journal

Read our guidance for manuscript revisions and related editorial policies: <https://link.springer.com/journal/44318/submission-guidelines#cms-Revised-submissions>

<https://media.springernature.com/original/springer-cms/rest/v1/content/27825798/data/v1>

- a point-by-point response to the referees' comments, with a detailed description of the changes made (as a word file).
- a word file of the manuscript text.
- individual production quality figure files (one file per figure)
- a complete author checklist
- Expanded View files (replacing Supplementary Information)
- a Reagents and Tools Table as part of the Methods section

Please remember: Digital image enhancement is acceptable practice, as long as it accurately represents the original data and conforms to community standards. If a figure has been subjected to significant electronic manipulation, this must be noted in the figure legend or in the 'Methods' section. The editors reserve the right to request original versions of figures and the original images that were used to assemble the figure.

We realize that it is difficult to revise to a specific deadline. In the interest of protecting the conceptual advance provided by the work, we recommend a revision within 3 months (1st Jun 2026). Please discuss the revision progress ahead of this time with the editor if you require more time to complete the revisions.

specific comments by editorial assistance team

- FUNDING: Please check whether or not the Austrian Academy of Sciences should be included in the list of funders in our system and add it in a separate line, if applicable.
- AUTHOR CONTRIBUTIONS: Please remove the list with author contributions from the manuscript text; they should be listed in our system only.
- DisclCIS: Please add the DECLARATION OF GENERATIVE AI AND AI-ASSISTED TECHNOLOGIES content to a general Declaration and Conflict of Interests Statement
- SOURCE DATA: completed checklist uploaded, images deposited to BioImage Archive (but not labeled by figure/panel). Fig 5D, Fig 6F, Fig 7C (blot/gel images) and Fig 6E (numerical data) have not been provided. Please upload the missing files to our system, as source data files and as one file per figure. Also, Fig EV2D, Fig EV3D and Fig EV4A have not been provided, you may wish to upload them as well (note that source data for EV figures is not mandatory).
- REAGENT TABLE: Please remove the table from the manuscript text and upload it as a separate document.
- SYNOPSIS IMAGE: Please upload a "synopsis image", which can be used as a visual summary of the main findings of the article. The image should be PNG or JPG format with pixel dimensions of 550 x 300 to 600 (width x height) and will display in the Synopsis section of the article page. It is important you follow exactly these instructions.
- SYNOPSIS TEXT: Please upload the "synopsis" in a separate document and as a related manuscript file. Please provide a short 'blurb' text summarising in two sentences the study (max. 250 characters), and three to four 'bullet points' highlighting the main findings of the study .
- FIGURE CALLOUTS: Please add citations for the panels of Figure EV3

Extra Notes:

- Please correct the order and headings of the manuscript sections to: Abstract / Introduction / Results / Discussion / Methods / Data Availability / Acknowledgements / Disclosure and Competing Interests Statement / References / Figure Legends / Tables / Expanded View Figure Legends
- Please delete the section "Resource Availability / Lead contact"

- Data Availability Section:

Please note that the specific URLs for S-BIAD2913, GSE305718 datasets are not provided in the data availability statement. Please correct.

- Figure legends:

1. Please note that the exact p values are not provided in the legend of figure EV4 A. Please provide.
2. Please indicate the statistical test used for data analysis in the legends of figures EV1 A, D
3. Please note that information related to n is missing in the legends of figures 6E, EV4 A. Please provide.
4. Please note that the error bars are not defined in the legend of figure 6E. Please correct.
5. Please note that the scale bar needs to be defined for figure 8B.
6. Please note that the dashed circles are not defined in the legend of figure 6G, EV2 E, EV3 E. This needs to be rectified.

specific comments by Referees

Referee #1:

The authors have done an excellent job addressing the comments, adding valuable new data and clarifications that substantially strengthen the manuscript.

There appears to be only one remaining point needed to ensure full transparency and reproducibility: the complete computational code should be made publicly available. At present, while the manuscript refers to prior work for computational methods, I was not able to locate a publicly accessible code package that reproduces the key analyses -particularly the quantitative "scores" used throughout (ping-pong and phasing scores), which are central to the manuscript's conclusions.

Depositing the full analysis pipeline in a stable, citable repository with a persistent DOI (e.g., Zenodo, Code Ocean, or an equivalent archive) would resolve this cleanly and substantially strengthen the long-term usability and impact of the work. Ideally, the dépôt would include key parameters and, where feasible, versioned dependencies or an environment specification.

Referee #2:

The authors have addressed all my concerns and have done a good job with it. It is now ready for publication.

Referee #3:

I loved this manuscript the first time and I still love it. Many of the other reviewer comments seem to be due to misunderstandings of the text and I think the authors did a very nice job of addressing those comments. I look forward to seeing this manuscript in print.

Responses to Reviewer Comments

“Retrovirus insertions in host transcripts trigger de novo piRNA immunity”

Baptiste Rafanel, Liudmila Protsenko, Dominik Handler, Julius Brennecke, Kirsten-André Senti

Reviewer comments are in **blue**.

Responses are in **black**.

Point by Point Responses to Reviewer Comments

Referee #1:

The authors have done an excellent job addressing the comments, adding valuable new data and clarifications that substantially strengthen the manuscript.

There appears to be only one remaining point needed to ensure full transparency and reproducibility: the complete computational code should be made publicly available. At present, while the manuscript refers to prior work for computational methods, I was not able to locate a publicly accessible code package that reproduces the key analyses -particularly the quantitative "scores" used throughout (ping-pong and phasing scores), which are central to the manuscript's conclusions.

Depositing the full analysis pipeline in a stable, citable repository with a persistent DOI (e.g., Zenodo, Code Ocean, or an equivalent archive) would resolve this cleanly and substantially strengthen the long-term usability and impact of the work. Ideally, the dépôt would include key parameters and, where feasible, versioned dependencies or an environment specification.

We thank Reviewer #1 for their comments. We agree with the suggestion to publish the code in a stable format and have therefore deposited the analysis pipeline used for the small RNA analysis via Zenodo at <https://doi.org/10.5281/zenodo.18882240> and Github at <https://github.com/BrenneckeLab/AnnotationPipeline>. The respective code for the calculation of ping-pong and phasing scores can be found in `generate_TEBg.sh` (line 406-459) and the `ping-pong.R` script.

Referee #2:

The authors have addressed all my concerns and have done a good job with it. It is now ready for publication.

We thank Reviewer #2 for their comments and support.

Referee #3:

I loved this manuscript the first time and I still love it. Many of the other reviewer comments seem to be due to misunderstandings of the text and I think the authors did a very nice job of addressing those comments. I look forward to seeing this manuscript in print.

We thank Reviewer #3 for their comments and appreciation.

Dear Dr. Brennecke,

I am pleased to inform you that your manuscript has been accepted for publication in the EMBO Journal.

You may qualify for financial assistance for your publication charges - either via a Springer Nature fully open access agreement or an EMBO initiative. Check your eligibility: <https://link.springer.com/journal/44318/how-to-publish-with-us>

Yours sincerely,

Yehu Moran
Academic Editor
The EMBO Journal

Please note that it is The EMBO Journal policy for the transcript of the editorial process (containing referee reports and your response letters) to be published as an online supplement to each paper. If you should prefer removal of any referee-only figures included in the point-by-point response(s), e.g. because they may still be used for future publication or because they have been reproduced from published work by others, please do let us know immediately via response email.

More information is available here: <https://link.springer.com/partners/embo-press/editorial-policies#Peer%20review>